# GUARD: A Safe Reinforcement Learning Benchmark

**Weiye Zhao**
Robotics Institute
Carnegie Mellon University
weiyezha@andrew.cmu.edu

**Rui Chen**
Robotics Institute
Carnegie Mellon University
ruic3@andrew.cmu.edu

**Yifan Sun**
Robotics Institute
Carnegie Mellon University
yifansu2@andrew.cmu.edu

**Ruixuan Liu**
Robotics Institute
Carnegie Mellon University
ruixuanl@andrew.cmu.edu

**Tianhao Wei**
Robotics Institute
Carnegie Mellon University
twei2@andrew.cmu.edu

**Changliu Liu**
Robotics Institute
Carnegie Mellon University
cliu6@andrew.cmu.edu

## Abstract

Due to the trial-and-error nature, it is typically challenging to apply RL algorithms to safety-critical real-world applications, such as autonomous driving, human-robot interaction, robot manipulation, etc, where such errors are not tolerable. Recently, safe RL (*i.e.*, constrained RL) has emerged rapidly in the literature, in which the agents explore the environment while satisfying constraints. Due to the diversity of algorithms and tasks, it remains difficult to compare existing safe RL algorithms. To fill that gap, we introduce GUARD, a **G**eneralized **U**nified SA**fe **R**einforcement Learning **D**evelopment Benchmark. GUARD has several advantages compared to existing benchmarks. First, GUARD is a generalized benchmark with a wide variety of RL agents, tasks, and safety constraint specifications. Second, GUARD comprehensively covers state-of-the-art safe RL algorithms with self-contained implementations. Third, GUARD is highly customizable in tasks and algorithms. We present a comparison of state-of-the-art safe RL algorithms in various task settings using GUARD and establish baselines that future work can build on.

## 1 Introduction

Reinforcement learning (RL) has achieved tremendous success in many fields over the past decades. In RL tasks, the agent explores and interacts with the environment by trial and error, and improves its performance by maximizing the long-term reward signal. RL algorithms enable the development of intelligent agents that can achieve human-competitive performance in a wide variety of tasks, such as games [Mnih et al., 2013, Zhao et al., 2019a, Silver et al., 2018, OpenAI et al., 2019, Vinyals et al., 2019, Zhao et al., 2019b], manipulation [Popov et al., 2017, Zhao et al., 2022a, Chen et al., 2023, Agostinelli et al., 2019, Shek et al., 2022, Zhao et al., 2020a, Noren et al., 2021], autonomous driving [Isele et al., 2019, Kiran et al., 2022, Gu et al., 2022a], robotics [Kober et al., 2013, Brunke et al., 2022, Zhao et al., 2022b, 2020b, Sun et al., 2023, Cheng et al., 2019], and more. Despite their outstanding performance in maximizing rewards, recent works [Garcıa and Fernández, 2015, Gu et al., 2022b, Zhao et al., 2023] focus on the safety aspect of training and deploying RL algorithms due to the safety concern [Zhao et al., 2022c, He et al., 2023a, Wei et al., 2022] in real-world safety-critical applications, *e.g.*, human-robot interaction, autonomous driving, etc. As safe RL topics emerge in the literature [Zhao et al., 2021, 2023, He et al., 2023b], it is crucial to employ a standardized benchmark for comparing and evaluating the performance of various safe RL algorithms across

Submitted to the 37th Conference on Neural Information Processing Systems (NeurIPS 2023) Track on Datasets and Benchmarks. Do not distribute.

different applications, ensuring a reliable transition from theory to practice. A benchmark includes 1) algorithms for comparison; 2) environments to evaluate algorithms; 3) a set of evaluation metrics, etc. There are benchmarks for unconfined RL and some safe RL, but not comprehensive enough [Duan et al., 2016, Brockman et al., 2016, Ellenberger, 2018–2019, Yu et al., 2019, Osband et al., 2020, Tunyasuvunakool et al., 2020, Dulac-Arnold et al., 2020, Zhang et al., 2022a].

To create a robust safe RL benchmark, we identify three essential pillars. Firstly, the benchmark must be **generalized**, accommodating diverse agents, tasks, and safety constraints. Real-world applications involve various agent types (e.g., drones, robot arms) with distinct complexities, such as different control degrees-of-freedom (DOF) and interaction modes (e.g., 2D planar or 3D spatial motion). The performance of algorithms is influenced by several factors, including variations in robots (such as observation and action space dimensions), tasks (interactive or non-interactive, 2D or 3D), and safety constraints (number, trespassibility, movability, and motion space). Therefore, providing a comprehensive environment to test the generalizability of safe RL algorithms is crucial.

Secondly, the benchmark should be **unified**, overcoming discrepancies in experiment setups prevalent in the emerging safe RL literature. A unified platform ensures consistent evaluation of different algorithms in controlled environments, promoting reliable performance comparison. Lastly, the benchmark must be **extensible**, allowing researchers to integrate new algorithms and extend setups to address evolving challenges. Given the ongoing progress in safe RL, the benchmark should incorporate major existing works and adapt to advancements. By encompassing these pillars, the benchmark provides a solid foundation for addressing these open problems in safe RL research.

In light of the above-mentioned pillars, this paper introduces GUARD, a **G**eneralized **U**nified **SA**fe **R**einforcement Learning **D**evelopment Benchmark. In particular, GUARD is developed based upon the Safety Gym [Ray et al., 2019], SafeRL-Kit [Zhang et al., 2022a] and SpinningUp [Achiam, 2018]. Unlike existing benchmarks, GUARD pushes the boundary beyond the limit by significantly extending the algorithms in comparison , types of agents and tasks, and safety constraint specifications. The contributions of this paper are as follows:

1. **Generalized benchmark with a wide range of agents.** GUARD genuinely supports **11** different agents, covering the majority of real robot types.

2. **Generalized benchmark with a wide range of tasks.** GUARD genuinely supports **7** different task specifications, which can be combined to represent most real robot tasks.

3. **Generalized benchmark with a wide range of safety constraints.** GUARD genuinely supports **8** different safety constraint specifications. The included constraint options comprehensively cover the safety requirements that would encounter in real-world applications.

4. **Unified benchmarking platform with comprehensive coverage of safe RL algorithms.** Guard implements **8** state-of-the-art safe RL algorithms following a unified code structure.

5. **Highly customizable benchmarking platform.** GUARD features a modularized design that enables effortless customization of new testing suites with self-customizable agents, tasks, and constraints. The algorithms in GUARD are self-contained, with a consistent structure and independent implementations, ensuring clean code organization and eliminating dependencies between different algorithms. This self-contained structure greatly facilitates the seamless integration of new algorithms for further extensions.

## 2   Related Work

**Open-source Libraries for Reinforcement Learning Algorithms**   Open-source RL libraries are code bases that implement representative RL algorithms for efficient deployment and comparison. They often serve as backbones for developing new RL algorithms, greatly facilitating RL research. We divide existing libraries into two categories: (a) safety-oriented RL libraries that support safe RL algorithms, and (b) general RL libraries that do not. Among safety-oriented libraries, Safety Gym [Ray et al., 2019] is the most famous one with highly configurable tasks and constraints but only

supports three safe RL methods. SafeRL-Kit [Zhang et al., 2022a] supports five safe RL methods while missing some key methods such as CPO [Achiam et al., 2017a]. Bullet-Safety-Gym [Gronauer, 2022] supports CPO but is limited in overall safe RL support at totally four methods. Compared to the above libraries, our proposed GUARD doubles the support at eight methods in total, covering a wider spectrum of general safe RL research. General RL libraries, on the other hand, can be summarized according to their backend into PyTorch [Achiam, 2018, Weng et al., 2022, Raffin et al., 2021, Liang et al., 2018], Tensorflow [Dhariwal et al., 2017, Hill et al., 2018], Jax [Castro et al., 2018, Hoffman et al., 2020], and Keras [Plappert, 2016]. In particular, SpinningUp [Achiam, 2018] serves as the major backbone of our GUARD benchmark on the safety-agnostic RL portion.

**Benchmark Platform for Safe RL Algorithms**   To facilitate safe RL research, the benchmark platform should support a wide range of task objectives, constraints, and agent types. Among existing work, the most representative one is Safety Gym [Ray et al., 2019] which is highly configurable. However, Safety Gym is limited in agent types in that it does not support high-dimensional agents (e.g., drone and arm) and lacks tasks with complex interactions (e.g., chase and defense). Moreover, Safety Gym only supports naive contact dynamics (e.g., touch and snap) instead of more realistic cases (e.g., objects bouncing off upon contact) in contact-rich tasks. Safe Control Gym [Yuan et al., 2022] is another open-source platform that supports very simple dynamics (i.e., cartpole, 1D/2D quadrotors) and only supports navigation tasks. Finally, Bullet Safety Gym [Gronauer, 2022] provides high-fidelity agents, but the types of agents are limited, and they only consider navigation tasks. Compared to the above platforms, our GUARD supports a much wider range of task objectives (e.g., 3D reaching, chase and defense) with a much larger variety of eight agents including high-dimensional ones such as drones, arms, ants, and walkers.

## 3   Preliminaries

**Markov Decision Process**   An Markov Decision Process (MDP) is specified by a tuple $(\mathcal{S}, \mathcal{A}, \gamma, \mathcal{R}, P, \rho)$, where $\mathcal{S}$ is the state space, and $\mathcal{A}$ is the control space, $\mathcal{R} : \mathcal{S} \times \mathcal{A} \to \mathbb{R}$ is the reward function, $0 \leq \gamma < 1$ is the discount factor, $\rho : \mathcal{S} \to [0, 1]$ is the starting state distribution, and $P : \mathcal{S} \times \mathcal{A} \times \mathcal{S} \to [0, 1]$ is the transition probability function (where $P(s'|s, a)$ is the probability of transitioning to state $s'$ given that the previous state was $s$ and the agent took action $a$ at state $s$). A stationary policy $\pi : \mathcal{S} \to \mathcal{P}(\mathcal{A})$ is a map from states to a probability distribution over actions, with $\pi(a|s)$ denoting the probability of selecting action $a$ in state $s$. We denote the set of all stationary policies by $\Pi$. Suppose the policy is parameterized by $\theta$; policy search algorithms search for the optimal policy within a set $\Pi_\theta \subset \Pi$ of parameterized policies.

The solution of the MDP is a policy $\pi$ that maximizes the performance measure $\mathcal{J}(\pi)$ computed via the discounted sum of reward:

$$\mathcal{J}(\pi) = \mathbb{E}_{\tau \sim \pi} \left[ \sum_{t=0}^{\infty} \gamma^t \mathcal{R}(s_t, a_t, s_{t+1}) \right], \tag{1}$$

where $\tau = [s_0, a_0, s_1, \cdots]$ is the state and control trajectory, and $\tau \sim \pi$ is shorthand for that the distribution over trajectories depends on $\pi : s_0 \sim \mu, a_t \sim \pi(\cdot|s_t), s_{t+1} \sim P(\cdot|s_t, a_t)$. Let $R(\tau) \doteq \sum_{t=0}^{\infty} \gamma^t \mathcal{R}(s_t, a_t, s_{t+1})$ be the discounted return of a trajectory. We define the on-policy value function as $V^\pi(s) \doteq \mathbb{E}_{\tau \sim \pi}[R(\tau)|s_0 = s]$, the on-policy action-value function as $Q^\pi(s, a) \doteq \mathbb{E}_{\tau \sim \pi}[R(\tau)|s_0 = s, a_0 = a]$, and the advantage function as $A^\pi(s, a) \doteq Q^\pi(s, a) - V^\pi(s)$.

**Constrained Markov Decision Process**   A constrained Markov Decision Process (CMDP) is an MDP augmented with constraints that restrict the set of allowable policies. Specifically, CMDP introduces a set of cost functions, $C_1, C_2, \cdots, C_m$, where $C_i : \mathcal{S} \times \mathcal{A} \times \mathcal{S} \to \mathbb{R}$ maps the state action transition tuple into a cost value. Similar to (1), we denote $\mathcal{J}_{C_i}(\pi) = \mathbb{E}_{\tau \sim \pi}[\sum_{t=0}^{\infty} \gamma^t C_i(s_t, a_t, s_{t+1})]$ as the cost measure for policy $\pi$ with respect to the cost function $C_i$. Hence, the set of feasible stationary policies for CMDP is then defined as $\Pi_C = \{\pi \in \Pi \big| \forall i, \mathcal{J}_{C_i}(\pi) \leq d_i\}$, where $d_i \in \mathbb{R}$. In CMDP, the objective is to select a feasible stationary policy $\pi$ that maximizes the performance:

$\max_{\pi \in \Pi_\theta \cap \Pi_C} \mathcal{J}(\pi)$. Lastly, we define on-policy value, action-value, and advantage functions for the cost as $V_{C_i}^\pi$, $Q_{C_i}^\pi$ and $A_{C_i}^\pi$, which as analogous to $V^\pi$, $Q^\pi$, and $A^\pi$, with $C_i$ replacing $R$.

# 4 GUARD Safe RL Library

## 4.1 Overall Implementation

GUARD contains the latest methods that can achieve safe RL: (i) end-to-end safe RL algorithms including CPO [Achiam et al., 2017a], TRPO-Lagrangian [Bohez et al., 2019], TRPO-FAC [Ma et al., 2021], TRPO-IPO [Liu et al., 2020], and PCPO [Yang et al., 2020]; and (ii) hierarchical safe RL algorithms including TRPO-SL (TRPO-Safety Layer) [Dalal et al., 2018] and TRPO-USL (TRPO-Unrolling Safety Layer) [Zhang et al., 2022a]. We also include TRPO [Schulman et al., 2015] as an unconstrained RL baseline. Note that GUARD only considers model-free approaches which rely less on assumptions than model-based ones. We highlight the benefits of our algorithm implementations in GUARD:

- GUARD comprehensively covers a **wide range of algorithms** that enforce safety in both hierarchical and end-to-end structures. Hierarchical methods maintain a separate safety layer, while end-to-end methods solve the constrained learning problem as a whole.

- GUARD provides a **fair comparison among safety components** by equipping every algorithm with the same reward-oriented RL backbone (i.e., TRPO [Schulman et al., 2015]), implementation (i.e., MLP policies with [64, 64] hidden layers and tanh activation), and training procedures. Hence, all algorithms inherit the performance guarantee of TRPO.

- GUARD is implemented in PyTorch with a clean structure where every algorithm is self-contained, enabling **fast customization and development** of new safe RL algorithms. GUARD also comes with unified logging and plotting utilities which makes analysis easy.

## 4.2 Unconstrained RL

**TRPO** We include TRPO [Schulman et al., 2015] since it is state-of-the-art and several safe RL algorithms are based on it. TRPO is an unconstrained RL algorithm and only maximizes performance $\mathcal{J}$. The key idea behind TRPO is to iteratively update the policy within a local range (trust region) of the most recent version $\pi_k$. Mathematically, TRPO updates policy via

$$\pi_{k+1} = \underset{\pi \in \Pi_\theta}{\mathbf{argmax}}\, \mathcal{J}(\pi) \quad \mathbf{s.t.}\, \mathcal{D}_{KL}(\pi, \pi_k) \le \delta, \tag{2}$$

where $\mathcal{D}_{KL}$ is Kullback-Leibler (KL) divergence, $\delta > 0$ and the set $\{\pi \in \Pi_\theta : \mathcal{D}_{KL}(\pi, \pi_k) \le \delta\}$ is called the *trust region*. To solve (2), TRPO applies Taylor expansion to the objective and constraint at $\pi_k$ to the first and second order, respectively. That results in an approximate optimization with linear objective and quadratic constraints (LOQC). TRPO guarantees a worst-case performance degradation.

## 4.3 End-to-End Safe RL

**CPO** Constrained Policy Optimizaiton (CPO) [Achiam et al., 2017b] handles CMDP by extending TRPO. Similar to TRPO, CPO also performs local policy updates in a trust region. Different from TRPO, CPO additionally requires $\pi_{k+1}$ to be constrained by $\Pi_\theta \cap \Pi_C$. For practical implementation, CPO replaces the objective and constraints with surrogate functions (advantage functions), which can easily be estimated from samples collected on $\pi_k$, formally:

$$\pi_{k+1} = \underset{\pi \in \Pi_\theta}{\mathbf{argmax}}\, \underset{\substack{s \sim d^{\pi_k} \\ a \sim \pi}}{\mathbb{E}} [A^{\pi_k}(s, a)] \tag{3}$$

$$\mathbf{s.t.}\quad \mathcal{D}_{KL}(\pi, \pi_k) \le \delta, \quad \mathcal{J}_{C_i}(\pi_k) + \frac{1}{1 - \gamma} \underset{\substack{s \sim d^{\pi_k} \\ a \sim \pi}}{\mathbb{E}} \left[ A_{C_i}^{\pi_k}(s, a) \right] \le d_i, i = 1, \cdots, m.$$

where $d^{\pi_k} \doteq (1 - \gamma) \sum_{t=0}^{H} \gamma^t P(s_t = s | \pi_k)$ is the discounted state distribution. Following TRPO, CPO also performs Taylor expansion on the objective and constraints, resulting in a Linear Objective with Linear and Quadratic Constraints (LOLQC). CPO inherits the worst-case performance degradation guarantee from TRPO and has a worst-case cost violation guarantee.

**PCPO** Projection-based Constrained Policy Optimization (PCPO) [Yang et al., 2020] is proposed based on CPO, where PCPO first maximizes reward using a trust region optimization method without any constraints, then PCPO reconciles the constraint violation (if any) by projecting the policy back onto the constraint set. Policy update then follows an analytical solution:

$$\pi_{k+1} = \pi_k + \sqrt{\frac{2\delta}{g^\top H^{-1} g}} H^{-1} g - \max\left(0, \frac{\sqrt{\frac{2\delta}{g^\top H^{-1} g}} g_c^\top H^{-1} g + b}{g_c^\top L^{-1} g_c}\right) L^{-1} g_c \tag{4}$$

where $g_c$ is the gradient of the cost advantage function, $g$ is the gradient of the reward advantage function, $H$ is the Hessian of the KL divergence constraint, $b$ is the constraint violation of the policy $\pi_k$, $L = \mathcal{I}$ for $L_2$ norm projection, and $L = H$ for KL divergence projection. PCPO provides a lower bound on reward improvement and an upper bound on constraint violation.

**TRPO-Lagrangian** Lagrangian methods solve constrained optimization by transforming hard constraints into soft constraints in the form of penalties for violations. Given the objective $\mathcal{J}(\pi)$ and constraints $\{\mathcal{J}_{C_i}(\pi) \leq d_i\}_i$, TRPO-Lagrangian [Bohez et al., 2019] first constructs the dual problem

$$\max_{\forall i, \lambda_i \geq 0} \min_{\pi \in \Pi_\theta} -\mathcal{J}(\pi) + \sum_i \lambda_i(\mathcal{J}_{C_i}(\pi) - d_i). \tag{5}$$

The update of $\theta$ is done via a trust region update with the objective of (2) replaced by that of (5) while fixing $\lambda_i$. The update of $\lambda_i$ is done via standard gradient ascend. Note that TRPO-Lagrangian does not have a theoretical guarantee for constraint satisfaction.

**TRPO-FAC** Inspired by Lagrangian methods and aiming at enforcing state-wise constraints (e.g., preventing state from stepping into infeasible parts in the state space), Feasible Actor Critic (FAC) [Ma et al., 2021] introduces a multiplier (dual variable) network. Via an alternative update procedure similar to that for (5), TRPO-FAC solves the *statewise* Lagrangian objective:

$$\max_{\forall i, \xi_i} \min_{\pi \in \Pi_\theta} -\mathcal{J}(\pi) + \sum_i \mathbb{E}_{s \sim d^{\pi_k}} \left[\lambda_{\xi_i}(s)(\mathcal{J}_{C_i}(\pi) - d_i)\right], \tag{6}$$

where $\lambda_{\xi_i}(s)$ is a parameterized Lagrangian multiplier network and is parameterized by $\xi_i$ for the $i$-th constraint. Note that TRPO-FAC does not have a theoretical guarantee for constraint satisfaction.

**TRPO-IPO** TRPO-IPO [Liu et al., 2020] incorporates constraints by augmenting the optimization objective in (2) with logarithmic barrier functions, inspired by the interior-point method [Boyd and Vandenberghe, 2004]. Ideally, the augmented objective is $I(\mathcal{J}_{C_i}(\pi) - d_i) = 0$ if $\mathcal{J}_{C_i}(\pi) - d_i \leq 0$ or $-\infty$ otherwise. Intuitively, that enforces the constraints since the violation penalty would be $-\infty$. To make the objective differentiable, $I(\cdot)$ is approximated by $\phi(x) = \log(-x)/t$ where $t > 0$ is a hyperparameter. Then TRPO-IPO solves (2) with the objective replaced by $\mathcal{J}_{\mathrm{IPO}}(\pi) = \mathcal{J}(\pi) + \sum_i \phi(\mathcal{J}_{C_i}(x) - d_i)$. TRPO-IPO does not have theoretical guarantees for constraint satisfaction.

### 4.4 Hierarchical Safe RL

**Safety Layer** Safety Layer [Dalal et al., 2018], added on top of the original policy network, conducts a quadratic-programming-based constrained optimization to project reference action into the nearest safe action. Mathematically:

$$a_t^{safe} = \operatorname*{argmin}_a \frac{1}{2} \|a - a_t^{ref}\|^2 \quad \textbf{s.t.} \quad \forall i, \bar{g}_{\varphi_i}(s_t)^\top a + C_i(s_{t-1}, a_{t-1}, s_t) \leq d_i \tag{7}$$

where $a_t^{ref} \sim \pi_k(\cdot | s_t)$, and $\bar{g}_{\varphi_i}(s_t)^\top a_t + C_i(s_{t-1}, a_{t-1}, s_t) \approx C_i(s_t, a_t, s_{t+1})$ is a $\varphi$ parameterized linear model. If there's only one constraint, (7) has a closed-form solution.

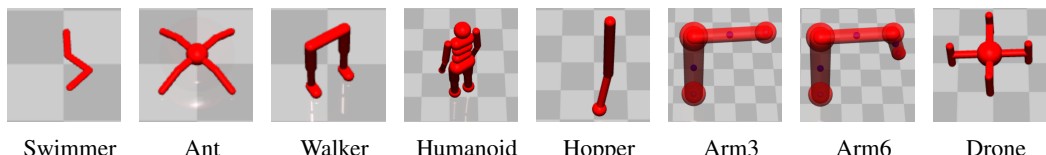

| Swimmer | Ant | Walker | Humanoid | Hopper | Arm3 | Arm6 | Drone |

Figure 1: Robots of our environments.

USL    Unrolling Safety Layer (USL) [Zhang et al., 2022b] is proposed to project the reference action into safe action via gradient-based correction. Specifically, USL iteratively updates the learned $Q_C(s, a)$ function with the samples collected during training. With step size $\eta$ and normalization factor $\mathcal{Z}$, USL performs gradient descent as $a_t^{safe} = a_t^{ref} - \frac{\eta}{\mathcal{Z}} \cdot \frac{\partial}{\partial a_t^{ref}} [Q_C(s_t, a_t^{ref}) - d]$.

## 5    GUARD Testing Suite

### 5.1    Robot Options

In GUARD testing suite, the agent (in the form of a robot) perceives the world through sensors and interacts with the world through actuators. Robots are specified through MuJoCo XML files. The suite is equipped with **8** types of pre-made robots that we use in our benchmark environments as whosn in Figure 1. The action space of the robots are continuous, and linearly scaled to [-1, +1].

**Swimmer** consist of three links and two joints. Each joint connects two links to form a linear chain. Swimmer can move around by applying **2** torques on the joints.

**Ant** is a quadrupedal robot composed a torso and four legs. Each of the four legs has a hip joint and a knee joint; and can move around by applying **8** torques to the joints.

**Walker** is a bipedal robot that consists of four main parts - a torso, two thighs, two legs, and two feet. Different from the knee joints and the ankle joints, each of the hip joints has three hinges in the $x$, $y$ and $z$ coordinates to help turning. With the torso height fixed, Walker can move around by controlling **10** joint torques.

**Humanoid** is also a bipedal robot that has a torso with a pair of legs and arms. Each leg of Humanoid consists of two joints (no ankle joint). Since we mainly focus on the navigation ability of the robots in designed tasks, the arm joints of Humanoid are fixed, which enables Humanoid to move around by only controlling **6** torques.

**Hopper** is a one-legged robot that consists of four main parts - a torso, a thigh, a leg, and a single foot. Similar to Walker, Hopper can move around by controlling **5** joint torques.

**Arm3** is designed to simulate a fixed three-joint robot arm. Arm is equipped with multiple sensors on each links in order to fully observe the environment. By controlling **3** joint torques, Arm can move its end effector around with high flexibility.

**Arm6** is designed to simulate a robot manipulator with a fixed base and six joints. Similar to Arm3, Arm6 can move its end effector around by controlling **6** torques.

**Drone** is designed to simulate a quadrotor. The interaction between the quadrotor and the air is simulated by applying four external forces on each of the propellers. The external forces are set to balance the gravity when the control action is zero. Drone can move in 3D space by applying **4** additional control forces on the propellers.

### 5.2    Task Options

We categorize robot tasks in two ways: (i) interactive versus non-interactive tasks, and (ii) 2D space versus 3D space tasks. 2D space tasks constrain agents to a planar space, while 3D space tasks do not. Non-interactive tasks primarily involve achieving a target state (e.g., trajectory tracking) while

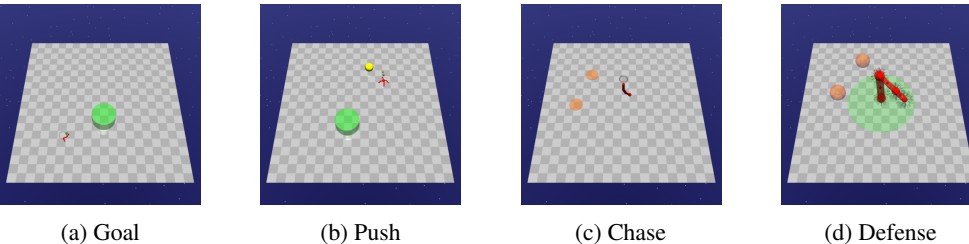

| (a) Goal | (b) Push | (c) Chase | (d) Defense |

Figure 2: Tasks of our environments.

interactive tasks (e.g., human-robot collaboration and unstructured object pickup) necessitate contact or non-contact interactions between the robot and humans or movable objects, rendering them more challenging. On a variety of tasks that cover different situations, GUARD facilitates a thorough evaluation of safe RL algorithms via the following tasks. See Table 17 for more information.

**Goal** (Figure 2a) requires the robot navigating towards a series of 2D or 3D goal positions. Upon reaching a goal, the location is randomly reset. The task provides a sparse reward upon goal achievement and a dense reward for making progress toward the goal.

**Push** (Figure 2b) requires the robot pushing a ball toward different goal positions. The task includes a sparse reward for the ball reaching the goal circle and a dense reward that encourages the agent to approach both the ball and the goal. Unlike pushing a box in Safety Gym, it is more challenging to push a ball since the ball can roll away and the contact dynamics are more complex.

**Chase** (Figure 2c) requires the robot tracking multiple dynamic targets. Those targets continuously move away from the robot at a slow speed. The dense reward component provides a bonus for minimizing the distance between the robot and the targets. The targets are constrained to a circular area. A 3D version of this task is also available, where the targets move within a restricted 3D space. Detailed dynamics of the targets is described in Appendix A.5.1.

**Defense** (Figure 2d) requires the robot to prevent dynamic targets from entering a protected circle area. The targets will head straight toward the protected area or avoid the robot if the robot gets too close. Dense reward component provides a bonus for increasing the cumulative distance between the targets and the protected area. Detailed dynamics of the targets is described in Appendix A.5.2.

## 5.3 Constraint Options

We classify constraints based on various factors: **trespassibility**: whether constraints are trespassable or untrespassable. Trespassable constraints allow violations without causing any changes to the robot's behaviors, and vice versa. (ii) **movability**: whether they are immovable, passively movable, or actively movable; and (iii) **motion space**: whether they pertain to 2D or 3D environments. To cover a comprehensive range of constraint configurations, we introduce additional constraint types via expanding Safety Gym. Please refer to Table 18 for all configurable constraints.

**3D Hazards** (Figure 3a) are dangerous 3D areas to avoid. These are floating spheres that are trespassable, and the robot is penalized for entering them.

**Ghosts** (Figure 3b) are dangerous areas to avoid. Different from hazards, ghosts always move toward the robot slowly, represented by circles on the ground. Ghosts can be either trespassable or untrespassable. The robot is penalized for touching the untrespassable ghosts and entering the trespassable ghosts. Moreover, ghosts can be configured to start chasing the robot when the distance from the robot is larger than some threshold. This feature together with the adjustable velocity allows users to design the ghosts with different aggressiveness. Detailed dynamics of the targets is described in Appendix A.5.3.

**3D Ghosts** (Figure 3c) are dangerous 3D areas to avoid. These are floating spheres as 3D version of ghosts, sharing the similar behaviour with ghosts.

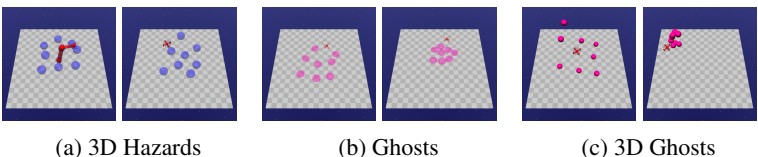

|           (a) 3D Hazards           |           (b) Ghosts           |           (c) 3D Ghosts           |

Figure 3: Constraints of our environments.

## 6 GUARD Experiments

**Benchmark Suite**   GUARD includes a set of predefined benchmark testing suite in form of `{Task}_{Robot}_{Constraint Number}{Constraint Type}`. The full list of our testing suite can be found in Table 20.

**Benchmark Results**   The summarized results can be found in Tables 21 to 25, and the learning rate curves are presented in Figures 6 to 10. As shown in Figure 4, we select 8 set of results to demonstrate the performance of different robot, task and constraints in GUARD. At a high level, the experiments show that all methods can consistently improve reward performance.

When comparing constrained RL methods to unconstrained RL methods, the former exhibit superior performance in terms of cost reduction. By incorporating constraints into the RL framework, the robot can navigate its environment while minimizing costs. This feature is particularly crucial for real-world applications where the avoidance of hazards and obstacles is of utmost importance. Nevertheless, it is important to point out that hierarchical RL methods (i.e., TRPO-SL and TRPO-USL) result in a trade-off between reward performance and cost reduction. While these methods excel at minimizing costs, they may sacrifice some degree of reward attainment in the process.

As shown in Figures 4b and 4c, tasks that involve high-dimensional robot action spaces and complex workspaces suffer from slower convergence due to the increased complexity of the learning problem. Moreover, the presence of dynamic ghosts in our tasks introduces further complexities. These tasks exhibit higher variance during the training process due to the collision-inducing behaviors of the dynamic ghosts. The robot must adapt and respond effectively to the ghosts' unpredictable movements. Addressing these challenges requires robust algorithms capable of handling the dynamic nature of the ghosts while optimizing the robot's overall performance. The influence of ghosts is evident by comparing Figure 4a and 4e, where the variance of cost performance increases with ghosts for several methods (e.g., PCPO and USL).

Figure 4a, 4f, 4g, and 4h illustrate the performance of a point robot on four distinct tasks. It is evident that the chase task exhibits the quickest convergence, while the defense task reveals the most performance gaps between methods. These verify that GUARD effectively benchmarks different methods under diverse scenarios.

## 7 Conclusions

Applying RL algorithms to safety-critical real-world applications poses significant challenges due to their trial-and-error nature. To address the problem, the literature has witnessed a rapid emergence of safe RL (constrained RL) approaches, where agents explore the environment while adhering to safety constraints. However, comparing diverse safe RL algorithms remains challenging. This paper introduces GUARD, the **G**eneralized **U**nified SA**f**e **R**einforcement Learning **D**evelopment Benchmark. GUARD offers several advantages over existing benchmarks. Firstly, it provides a generalized framework with a wide range of RL agents, tasks, and constraint specifications. Secondly, GUARD has self-contained implementations of a comprehensive range of state-of-the-art safe RL algorithms. Lastly, GUARD is highly customizable, allowing researchers to tailor tasks and algorithms to specific needs. Using GUARD, we present a comparative analysis of state-of-the-art safe RL algorithms across various task settings, establishing essential baselines for future research.

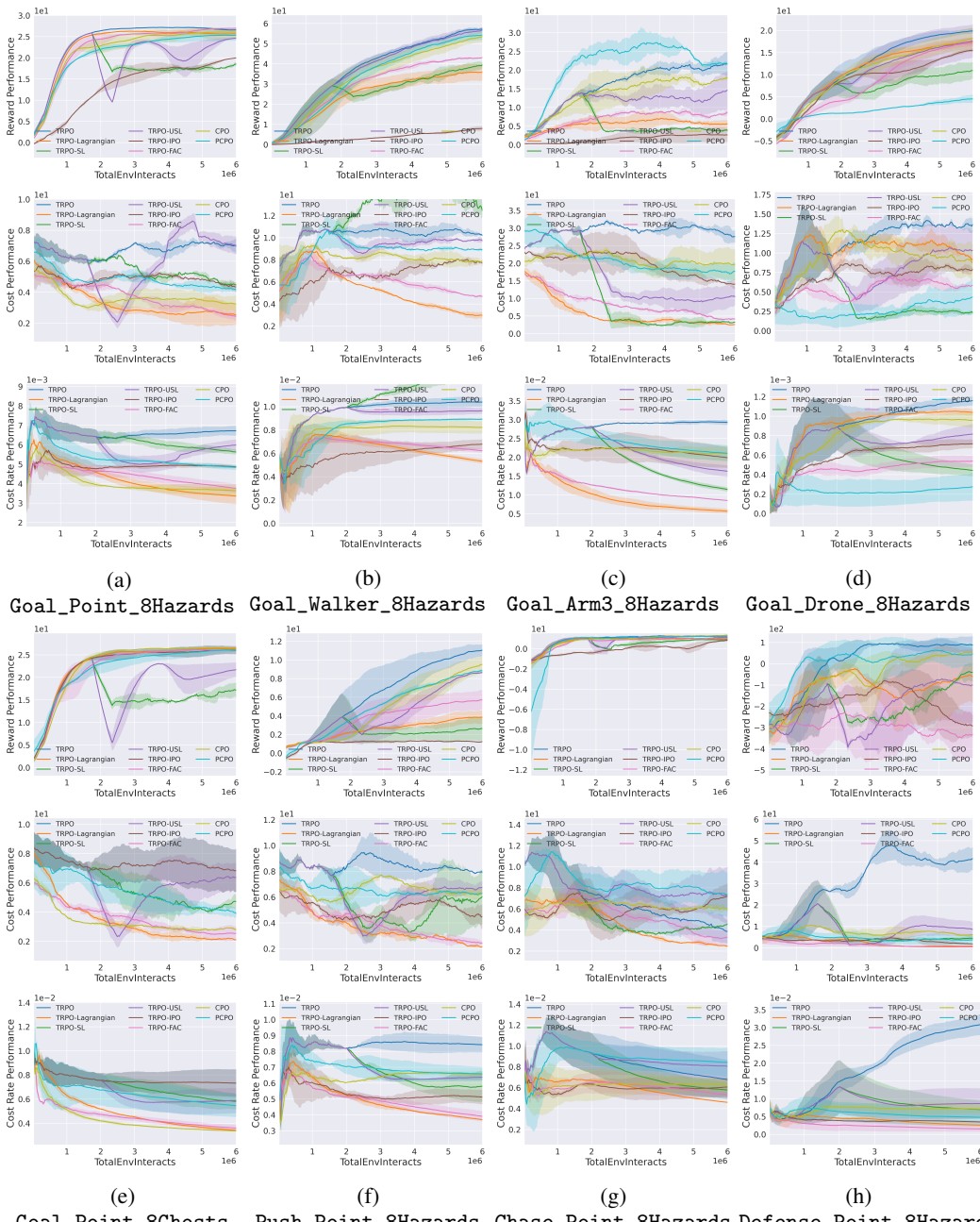

Figure 4: Comparison of results from four representative tasks. (a) to (d) cover four robots on the goal task. (e) shows the performance of a task with ghosts. (f) to (h) cover three different tasks with the point robot.

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
