# A    Environment Details

## A.1    Observation Space and Action space of different robots

The action space and observation space of different robots are summarized in Tables 1 to 16

Table 1: Action space of Swimmer

| Num | Action | Min | Max | Name in XML | Joint | Unit |
| --- | --- | --- | --- | --- | --- | --- |
| 0 | Torque applied on the first rotor | -1 | 1 | motor1_rot | hinge | torque ($Nm$) |
| 1 | Torque applied on the second rotor | -1 | 1 | motor2_rot | hinge | torque ($Nm$) |

Table 2: Observation space of Swimmer

| Num | Observation | Min | Max | Name in XML | Joint | Unit |
| --- | --- | --- | --- | --- | --- | --- |
| 0/1/2 | 3-axis linear acceleration of the torso (including gravity) | -Inf | Inf | accelerometer | free | acceleration ($m/s^2$) |
| 3/4/5 | 3-axis linear velocity of the torso | -Inf | Inf | velocimeter | free | velocity ($m/s$) |
| 6/7/8 | 3-axis angular velocity of the torso | -Inf | Inf | gyro | free | angular velocity ($rad/s$) |
| 9/10/11 | 3D magnetic flux vector at the torso | -Inf | Inf | magnetometer | free | magnetic flux density ($T$) |
| 12 | Contact force at the first tip | 0 | Inf | touch_point1 | - | force($N$) |
| 13 | Contact force at the first rotor | 0 | Inf | touch_point2 | - | force($N$) |
| 14 | Contact force at the second rotor | 0 | Inf | touch_point3 | - | force($N$) |
| 15 | Contact force at the second tip | 0 | Inf | touch_point4 | - | force($N$) |
| 16 | Angle of the first rotor | -Inf | Inf | jointpos_motor1_rot | hinge | angle ($rad$) |
| 17 | Angle of the second rotor | -Inf | Inf | jointpos_motor2_rot | hinge | angle ($rad$) |
| 18 | Angular velocity of the first rotor | -Inf | Inf | jointvel_motor1_rot | hinge | angular velocity ($rad/s$) |
| 19 | Angular velocity of the second rotor | -Inf | Inf | jointvel_motor2_rot | hinge | angular velocity ($rad/s$) |

Table 3: Action space of Ant

| Num | Action | Min | Max | Name in XML | Joint | Unit |
| --- | --- | --- | --- | --- | --- | --- |
| 0 | Torque applied on the rotor between the torso and front left hip | -1 | 1 | hip_1 | hinge | torque $(Nm)$ |
| 1 | Torque applied on the rotor between the front left two links | -1 | 1 | ankle_1 | hinge | torque $(Nm)$ |
| 2 | Torque applied on the rotor between the torso and front right hip | -1 | 1 | hip_2 | hinge | torque $(Nm)$ |
| 3 | Torque applied on the rotor between the front right two links | -1 | 1 | ankle_2 | hinge | torque $(Nm)$ |
| 4 | Torque applied on the rotor between the torso and back left hip | -1 | 1 | hip_3 | hinge | torque $(Nm)$ |
| 5 | Torque applied on the rotor between the back left two links | -1 | 1 | ankle_3 | hinge | torque $(Nm)$ |
| 6 | Torque applied on the rotor between the torso and back right hip | -1 | 1 | hip_4 | hinge | torque $(Nm)$ |
| 7 | Torque applied on the rotor between the back right two links | -1 | 1 | ankle_4 | hinge | torque $(Nm)$ |

Table 4: Observation space of Ant

| Num | Observation | Min | Max | Name in XML | Joint | Unit |
| --- | --- | --- | --- | --- | --- | --- |
| 0/1/2 | 3-axis linear acceleration of the torso (including gravity) | -Inf | Inf | accelerometer | free | acceleration $(m/s^2)$ |
| 3/4/5 | 3-axis linear velocity of the torso | -Inf | Inf | velocimeter | free | velocity $(m/s)$ |
| 6/7/8 | 3-axis angular velocity velocity of the torso | -Inf | Inf | gyro | free | angular velocity $(rad/s)$ |
| 9/10/11 | 3D magnetic flux vector at the torso | -Inf | Inf | magnetometer | free | magnetic flux density $(T)$ |
| 12 | Contact force at the front left ankle | 0 | Inf | touch_ankle_1a | - | force$(N)$ |
| 13 | Contact force at the front right ankle | 0 | Inf | touch_ankle_2a | - | force$(N)$ |
| 14 | Contact force at the back left ankle | 0 | Inf | touch_ankle_3a | - | force$(N)$ |
| 15 | Contact force at the back left ankle | 0 | Inf | touch_ankle_4a | - | force$(N)$ |
| 16 | Contact force at the end of the front left leg | 0 | Inf | touch_ankle_1b | - | force$(N)$ |
| 17 | Contact force at the end of the front right leg | 0 | Inf | touch_ankle_2b | - | force$(N)$ |
| 18 | Contact force at the end of the back left leg | 0 | Inf | touch_ankle_3b | - | force$(N)$ |
| 19 | Contact force at the end of the back left leg | 0 | Inf | touch_ankle_4b | - | force$(N)$ |
| 20 | Angle of the front left hip | -Inf | Inf | jointpos_hip_1 | hinge | angle $(rad)$ |
| 21 | Angle of the front right hip | -Inf | Inf | jointpos_hip_2 | hinge | angle $(rad)$ |
| 22 | Angle of the back left hip | -Inf | Inf | jointpos_hip_3 | hinge | angle $(rad)$ |
| 23 | Angle of the back right hip | -Inf | Inf | jointpos_hip_4 | hinge | angle $(rad)$ |
| 24 | Angle of the front left ankle | -Inf | Inf | jointpos_ankle_1 | hinge | angle $(rad)$ |
| 25 | Angle of the front right ankle | -Inf | Inf | jointpos_ankle_2 | hinge | angle $(rad)$ |
| 26 | Angle of the back left ankle | -Inf | Inf | jointpos_ankle_3 | hinge | angle $(rad)$ |
| 27 | Angle of the back right ankle | -Inf | Inf | jointpos_ankle_4 | hinge | angle $(rad)$ |
| 28 | Angular velocity of the front left hip | -Inf | Inf | jointvel_hip_1 | hinge | angular velocity $(rad/s)$ |
| 29 | Angular velocity of the front right hip | -Inf | Inf | jointvel_hip_2 | hinge | angular velocity $(rad/s)$ |
| 30 | Angular velocity of the back left hip | -Inf | Inf | jointvel_hip_3 | hinge | angular velocity $(rad/s)$ |
| 31 | Angular velocity of the back right hip | -Inf | Inf | jointvel_hip_4 | hinge | angular velocity $(rad/s)$ |
| 32 | Angular velocity of the front left ankle | -Inf | Inf | jointvel_ankle_1 | hinge | angular velocity $(rad/s)$ |
| 33 | Angular velocity of the front right ankle | -Inf | Inf | jointvel_ankle_2 | hinge | angular velocity $(rad/s)$ |
| 34 | Angular velocity of the back left ankle | -Inf | Inf | jointvel_ankle_3 | hinge | angular velocity $(rad/s)$ |
| 35 | Angular velocity of the back right ankle | -Inf | Inf | jointvel_ankle_4 | hinge | angular velocity $(rad/s)$ |

Table 5: Action space of Walker

| Num | Action | Min | Max | Name in XML | Joint | Unit |
| --- | --- | --- | --- | --- | --- | --- |
| 0 | Torque applied on the rotor between torso and the right hip (x-coordinate) | -1 | 1 | right_hip_x | hinge | torque $(Nm)$ |
| 1 | Torque applied on the rotor between torso and the right hip (z-coordinate) | -1 | 1 | right_hip_z | hinge | torque $(Nm)$ |
| 2 | Torque applied on the rotor between torso and the right hip (y-coordinate) | -1 | 1 | right_hip_y | hinge | torque $(Nm)$ |
| 3 | Torque applied on the right leg rotor | -1 | 1 | right_leg_joint | hinge | torque $(Nm)$ |
| 4 | Torque applied on the right foot rotor | -1 | 1 | right_foot_joint | hinge | torque $(Nm)$ |
| 5 | Torque applied on the rotor between torso and the left hip (x-coordinate) | -1 | 1 | left_hip_x | hinge | torque $(Nm)$ |
| 6 | Torque applied on the rotor between torso and the left hip (z-coordinate) | -1 | 1 | left_hip_z | hinge | torque $(Nm)$ |
| 7 | Torque applied on the rotor between torso and the left hip (y-coordinate) | -1 | 1 | left_hip_y | hinge | torque $(Nm)$ |
| 8 | Torque applied on the left leg rotor | -1 | 1 | left_leg_joint | hinge | torque $(Nm)$ |
| 9 | Torque applied on the left foot rotor | -1 | 1 | left_foot_joint | hinge | torque $(Nm)$ |

Table 6: Observation space of Walker

| Num | Observation | Min | Max | Name in XML | Joint | Unit |
| --- | --- | --- | --- | --- | --- | --- |
| 0/1/2 | 3-axis linear acceleration of the torso (including gravity) | -Inf | Inf | accelerometer | free | acceleration $(m/s^2)$ |
| 3/4/5 | 3-axis linear velocity of the torso | -Inf | Inf | velocimeter | free | velocity $(m/s)$ |
| 6/7/8 | 3-axis angular velocity of the torso | -Inf | Inf | gyro | free | angular velocity $(rad/s)$ |
| 9/10/11 | 3D magnetic flux vector at the torso | -Inf | Inf | magnetometer | free | magnetic flux density $(T)$ |
| 12 | Contact force at the right foot | 0 | Inf | touch_right_foot | - | force$(N)$ |
| 13 | Contact force at the left foot | 0 | Inf | touch_left_foot | - | force$(N)$ |
| 14 | Angle of the right hip (x-coordinate) | -Inf | Inf | jointpos_right_hip_x | hinge | angle $(rad)$ |
| 15 | Angle of the right hip (z-coordinate) | -Inf | Inf | jointpos_right_hip_z | hinge | angle $(rad)$ |
| 16 | Angle of the right hip (y-coordinate) | -Inf | Inf | jointpos_right_hip_y | hinge | angle $(rad)$ |
| 17 | Angle of the right leg | -Inf | Inf | jointpos_right_leg | hinge | angle $(rad)$ |
| 18 | Angle of the right foot | -Inf | Inf | jointpos_right_foot | hinge | angle $(rad)$ |
| 19 | Angle of the left hip (x-coordinate) | -Inf | Inf | jointpos_left_hip_x | hinge | angle $(rad)$ |
| 20 | Angle of the left hip (z-coordinate) | -Inf | Inf | jointpos_left_hip_z | hinge | angle $(rad)$ |
| 21 | Angle of the left hip (y-coordinate) | -Inf | Inf | jointpos_left_hip_y | hinge | angle $(rad)$ |
| 22 | Angle of the left leg | -Inf | Inf | jointpos_left_leg | hinge | angle $(rad)$ |
| 23 | Angle of the left foot | -Inf | Inf | jointpos_left_foot | hinge | angle $(rad)$ |
| 24 | Angular velocity of the right hip (x-coordinate) | -Inf | Inf | jointvel_right_hip_x | hinge | angular velocity $(rad/s)$ |
| 25 | Angular velocity of the right hip (z-coordinate) | -Inf | Inf | jointvel_right_hip_z | hinge | angular velocity $(rad/s)$ |
| 26 | Angular velocity of the right hip (y-coordinate) | -Inf | Inf | jointvel_right_hip_y | hinge | angular velocity $(rad/s)$ |
| 27 | Angular velocity of the right leg | -Inf | Inf | jointvel_right_leg | hinge | angular velocity $(rad/s)$ |
| 28 | Angular velocity of the right foot | -Inf | Inf | jointvel_right_foot | hinge | angular velocity $(rad/s)$ |
| 29 | Angular velocity of the left hip (x-coordinate) | -Inf | Inf | jointvel_left_hip_x | hinge | angular velocity $(rad/s)$ |
| 30 | Angular velocity of the left hip (z-coordinate) | -Inf | Inf | jointvel_left_hip_z | hinge | angular velocity $(rad/s)$ |
| 31 | Angular velocity of the left hip (y-coordinate) | -Inf | Inf | jointvel_left_hip_y | hinge | angular velocity $(rad/s)$ |
| 32 | Angular velocity of the left leg | -Inf | Inf | jointvel_left_leg | hinge | angular velocity $(rad/s)$ |
| 33 | Angular velocity of the left foot | -Inf | Inf | jointvel_left_foot | hinge | angular velocity $(rad/s)$ |

Table 7: Action space of Humanoid

| Num | Action | Min | Max | Name in XML | Joint | Unit |
|---|---|---|---|---|---|---|
| 0 | Torque applied on the rotor between torso and the right hip (x-coordinate) | -1 | 1 | right_hip_x | hinge | torque ($Nm$) |
| 1 | Torque applied on the rotor between torso and the right hip (z-coordinate) | -1 | 1 | right_hip_z | hinge | torque ($Nm$) |
| 2 | Torque applied on the rotor between torso and the right hip (y-coordinate) | -1 | 1 | right_hip_y | hinge | torque ($Nm$) |
| 3 | Torque applied on the right knee rotor | -1 | 1 | right_knee | hinge | torque ($Nm$) |
| 4 | Torque applied on the rotor between torso and the left hip (x-coordinate) | -1 | 1 | left_hip_x | hinge | torque ($Nm$) |
| 5 | Torque applied on the rotor between torso and the left hip (z-coordinate) | -1 | 1 | left_hip_z | hinge | torque ($Nm$) |
| 6 | Torque applied on the rotor between torso and the left hip (y-coordinate) | -1 | 1 | left_hip_y | hinge | torque ($Nm$) |
| 7 | Torque applied on the left knee rotor | -1 | 1 | left_knee | hinge | torque ($Nm$) |

Table 8: Observation space of Humanoid

| Num | Observation | Min | Max | Name in XML | Joint | Unit |
|---|---|---|---|---|---|---|
| 0/1/2 | 3-axis linear acceleration of the torso (including gravity) | -Inf | Inf | accelerometer | free | acceleration ($m/s^2$) |
| 3/4/5 | 3-axis linear velocity of the torso | -Inf | Inf | velocimeter | free | velocity ($m/s$) |
| 6/7/8 | 3-axis angular velocity of the torso | -Inf | Inf | gyro | free | angular velocity ($rad/s$) |
| 9/10/11 | 3D magnetic flux vector at the torso | -Inf | Inf | magnetometer | free | magnetic flux density ($T$) |
| 12 | Contact force at the right foot | 0 | Inf | touch_right_foot | - | force($N$) |
| 13 | Contact force at the left foot | 0 | Inf | touch_left_foot | - | force($N$) |
| 14 | Angle of the right hip (x-coordinate) | -Inf | Inf | jointpos_right_hip_x | hinge | angle ($rad$) |
| 15 | Angle of the right hip (z-coordinate) | -Inf | Inf | jointpos_right_hip_z | hinge | angle ($rad$) |
| 16 | Angle of the right hip (y-coordinate) | -Inf | Inf | jointpos_right_hip_y | hinge | angle ($rad$) |
| 17 | Angle of the right knee | -Inf | Inf | jointpos_right_knee | hinge | angle ($rad$) |
| 18 | Angle of the left hip (x-coordinate) | -Inf | Inf | jointpos_left_hip_x | hinge | angle ($rad$) |
| 19 | Angle of the left hip (z-coordinate) | -Inf | Inf | jointpos_left_hip_z | hinge | angle ($rad$) |
| 20 | Angle of the left hip (y-coordinate) | -Inf | Inf | jointpos_left_hip_y | hinge | angle ($rad$) |
| 21 | Angle of the left leg | -Inf | Inf | jointpos_left_knee | hinge | angle ($rad$) |
| 22 | Angular velocity of the right hip (x-coordinate) | -Inf | Inf | jointvel_right_hip_x | hinge | angular velocity ($rad/s$) |
| 23 | Angular velocity of the right hip (z-coordinate) | -Inf | Inf | jointvel_right_hip_z | hinge | angular velocity ($rad/s$) |
| 24 | Angular velocity of the right hip (y-coordinate) | -Inf | Inf | jointvel_right_hip_y | hinge | angular velocity ($rad/s$) |
| 25 | Angular velocity of the right knee | -Inf | Inf | jointvel_right_knee | hinge | angular velocity ($rad/s$) |
| 26 | Angular velocity of the left hip (x-coordinate) | -Inf | Inf | jointvel_left_hip_x | hinge | angular velocity ($rad/s$) |
| 27 | Angular velocity of the left hip (z-coordinate) | -Inf | Inf | jointvel_left_hip_z | hinge | angular velocity ($rad/s$) |
| 28 | Angular velocity of the left hip (y-coordinate) | -Inf | Inf | jointvel_left_hip_y | hinge | angular velocity ($rad/s$) |
| 29 | Angular velocity of the left knee | -Inf | Inf | jointvel_left_knee | hinge | angular velocity ($rad/s$) |

Table 9: Action space of Hopper

| Num | Action | Min | Max | Name in XML | Joint | Unit |
| --- | --- | --- | --- | --- | --- | --- |
| 0 | Torque applied on the rotor between torso and the hip (x-coordinate) | -1 | 1 | hip_x | hinge | torque $(Nm)$ |
| 1 | Torque applied on the rotor between torso and the hip (z-coordinate) | -1 | 1 | hip_z | hinge | torque $(Nm)$ |
| 2 | Torque applied on the rotor between torso and the hip (y-coordinate) | -1 | 1 | hip_y | hinge | torque $(Nm)$ |
| 3 | Torque applied on the thigh rotor | -1 | 1 | thigh_joint | hinge | torque $(Nm)$ |
| 4 | Torque applied on the leg rotor | -1 | 1 | leg_joint | hinge | torque $(Nm)$ |
| 5 | Torque applied on the foot rotor | -1 | 1 | foot_joint | hinge | torque $(Nm)$ |

Table 10: Observation space of Hopper

| Num | Observation | Min | Max | Name in XML | Joint | Unit |
| --- | --- | --- | --- | --- | --- | --- |
| 0/1/2 | 3-axis linear acceleration of the torso (including gravity) | -Inf | Inf | accelerometer | free | acceleration $(m/s^2)$ |
| 3/4/5 | 3-axis linear velocity of the torso | -Inf | Inf | velocimeter | free | velocity $(m/s)$ |
| 6/7/8 | 3-axis angular velocity of the torso | -Inf | Inf | gyro | free | angular velocity $(rad/s)$ |
| 9/10/11 | 3D magnetic flux vector at the torso | -Inf | Inf | magnetometer | free | magnetic flux density $(T)$ |
| 12 | Contact force at the foot | 0 | Inf | touch_foot | - | force$(N)$ |
| 13 | Angle of the hip (x-coordinate) | -Inf | Inf | jointpos_hip_x | hinge | angle $(rad)$ |
| 14 | Angle of the hip (z-coordinate) | -Inf | Inf | jointpos_hip_z | hinge | angle $(rad)$ |
| 15 | Angle of the hip (y-coordinate) | -Inf | Inf | jointpos_hip_y | hinge | angle $(rad)$ |
| 16 | Angle of the thigh | -Inf | Inf | jointpos_thigh | hinge | angle $(rad)$ |
| 17 | Angle of the leg | -Inf | Inf | jointpos_leg | hinge | angle $(rad)$ |
| 18 | Angle of the foot | -Inf | Inf | jointpos_foot | hinge | angle $(rad)$ |
| 19 | Angular velocity of the hip (x-coordinate) | -Inf | Inf | jointvel_hip_x | hinge | angular velocity $(rad/s)$ |
| 20 | Angular velocity of the hip (z-coordinate) | -Inf | Inf | jointvel_hip_z | hinge | angular velocity $(rad/s)$ |
| 21 | Angular velocity of the hip (y-coordinate) | -Inf | Inf | jointvel_hip_y | hinge | angular velocity $(rad/s)$ |
| 22 | Angular velocity of the thigh | -Inf | Inf | jointvel_thigh | hinge | angular velocity $(rad/s)$ |
| 23 | Angular velocity of the leg | -Inf | Inf | jointvel_leg | hinge | angular velocity $(rad/s)$ |
| 24 | Angular velocity of the foot | -Inf | Inf | jointvel_foot | hinge | angular velocity $(rad/s)$ |

Table 11: Action space of Arm3

| Num | Action | Min | Max | Name in XML | Joint | Unit |
|---|---|---|---|---|---|---|
| 0 | Torque applied on the first joint (connecting the base point and the first link) | -1 | 1 | joint_1 | hinge | torque $(Nm)$ |
| 1 | Torque applied on the second joint(connecting the first and the second link) | -1 | 1 | joint_2 | hinge | torque $(Nm)$ |
| 2 | Torque applied on the third joint (connecting the second and the third link) | -1 | 1 | joint_3 | hinge | torque $(Nm)$ |

Table 12: Observation space of Arm3

| Num | Observation | Min | Max | Name in XML | Joint | Unit |
|---|---|---|---|---|---|---|
| 0/1/2 | 3-axis linear acceleration of the first link (including gravity) | -Inf | Inf | accelerometer_link_1 | free | acceleration $(m/s^2)$ |
| 3/4/5 | 3-axis linear velocity of the first link | -Inf | Inf | velocimeter_link_1 | free | velocity $(m/s)$ |
| 6/7/8 | 3-axis angular velocity velocity of the first link | -Inf | Inf | gyro_link_1 | free | angular velocity $(rad/s)$ |
| 9/10/11 | 3D magnetic flux vector at the first link | -Inf | Inf | magnetometer_link_1 | free | magnetic flux density $(T)$ |
| 12/13/14 | 3-axis linear acceleration of the second link (including gravity) | -Inf | Inf | accelerometer_link_2 | free | acceleration $(m/s^2)$ |
| 15/16/17 | 3-axis linear velocity of the second link | -Inf | Inf | velocimeter_link_2 | free | velocity $(m/s)$ |
| 18/19/20 | 3-axis angular velocity velocity of the second link | -Inf | Inf | gyro_link_2 | free | angular velocity $(rad/s)$ |
| 21/22/23 | 3D magnetic flux vector at the second link | -Inf | Inf | magnetometer_link_2 | free | magnetic flux density $(T)$ |
| 24/25/26 | 3-axis linear acceleration of the third link (including gravity) | -Inf | Inf | accelerometer_link_3 | free | acceleration $(m/s^2)$ |
| 27/28/29 | 3-axis linear velocity of the third link | -Inf | Inf | velocimeter_link_3 | free | velocity $(m/s)$ |
| 30/31/32 | 3-axis angular velocity velocity of the third link | -Inf | Inf | gyro_link_3 | free | angular velocity $(rad/s)$ |
| 33/34/35 | 3D magnetic flux vector at the third link | -Inf | Inf | magnetometer_link_3 | free | magnetic flux density $(T)$ |
| 36/37/38 | 3-axis linear acceleration of the fourth link (including gravity) | -Inf | Inf | accelerometer_link_4 | free | acceleration $(m/s^2)$ |
| 39/40/41 | 3-axis linear velocity of the fourth link | -Inf | Inf | velocimeter_link_4 | free | velocity $(m/s)$ |
| 42/43/44 | 3-axis angular velocity velocity of the fourth link | -Inf | Inf | gyro_link_4 | free | angular velocity $(rad/s)$ |
| 45/46/47 | 3D magnetic flux vector at the fourth link | -Inf | Inf | magnetometer_link_4 | free | magnetic flux density $(T)$ |
| 48/49/50 | 3-axis linear acceleration of the fifth link (including gravity) | -Inf | Inf | accelerometer_link_5 | free | acceleration $(m/s^2)$ |
| 51/52/53 | 3-axis linear velocity of the fifth link | -Inf | Inf | velocimeter_link_5 | free | velocity $(m/s)$ |
| 54/55/56 | 3-axis angular velocity velocity of the fifth link | -Inf | Inf | gyro_link_5 | free | angular velocity $(rad/s)$ |
| 57/58/59 | 3D magnetic flux vector at the fifth link | -Inf | Inf | magnetometer_link_5 | free | magnetic flux density $(T)$ |
| 60 | Angle of the first joint | -Inf | Inf | jointpos_joint_1 | hinge | angle $(rad)$ |
| 61 | Angle of the second joint | -Inf | Inf | jointpos_joint_2 | hinge | angle $(rad)$ |
| 62 | Angle of the third joint | -Inf | Inf | jointpos_joint_3 | hinge | angle $(rad)$ |
| 63 | Angular velocity of the first joint | -Inf | Inf | jointvel_joint_1 | hinge | angular velocity $(rad/s)$ |
| 64 | Angular velocity of the second joint | -Inf | Inf | jointvel_joint_2 | hinge | angular velocity $(rad/s)$ |
| 65 | Angular velocity of the third joint | -Inf | Inf | jointvel_joint_3 | hinge | angular velocity $(rad/s)$ |
| 66 | Contact force at the end effector | 0 | Inf | touch_end_effector | - | force$(N)$ |

Table 13: Action space of Arm6

| Num | Action | Min | Max | Name in XML | Joint | Unit |
|---|---|---|---|---|---|---|
| 0 | Torque applied on the first joint (connecting the base point and the first link) | -1 | 1 | joint_1 | hinge | torque ($Nm$) |
| 1 | Torque applied on the second joint(connecting the first and the second link) | -1 | 1 | joint_2 | hinge | torque ($Nm$) |
| 2 | Torque applied on the third joint (connecting the second and the third link) | -1 | 1 | joint_3 | hinge | torque ($Nm$) |
| 3 | Torque applied on the fourth joint (connecting the third and the fourth link) | -1 | 1 | joint_4 | hinge | torque ($Nm$) |
| 4 | Torque applied on the fifth joint (connecting the fourth and the fifth link) | -1 | 1 | joint_5 | hinge | torque ($Nm$) |
| 5 | Torque applied on the sixth joint (connecting the fifth and the sixth link) | -1 | 1 | joint_6 | hinge | torque ($Nm$) |

Table 14: Observation space of Arm6

| Num | Observation | Min | Max | Name in XML | Joint | Unit |
|---|---|---|---|---|---|---|
| 0/1/2 | 3-axis linear acceleration of the first link (including gravity) | -Inf | Inf | accelerometer_link_1 | free | acceleration $(m/s^2)$ |
| 3/4/5 | 3-axis linear velocity of the first link | -Inf | Inf | velocimeter_link_1 | free | velocity $(m/s)$ |
| 6/7/8 | 3-axis angular velocity velocity of the first link | -Inf | Inf | gyro_link_1 | free | angular velocity $(rad/s)$ |
| 9/10/11 | 3D magnetic flux vector at the first link | -Inf | Inf | magnetometer_link_1 | free | magnetic flux density $(T)$ |
| 12/13/14 | 3-axis linear acceleration of the second link (including gravity) | -Inf | Inf | accelerometer_link_2 | free | acceleration $(m/s^2)$ |
| 15/16/17 | 3-axis linear velocity of the second link | -Inf | Inf | velocimeter_link_2 | free | velocity $(m/s)$ |
| 18/19/20 | 3-axis angular velocity velocity of the second link | -Inf | Inf | gyro_link_2 | free | angular velocity $(rad/s)$ |
| 21/22/23 | 3D magnetic flux vector at the second link | -Inf | Inf | magnetometer_link_2 | free | magnetic flux density $(T)$ |
| 24/25/26 | 3-axis linear acceleration of the third link (including gravity) | -Inf | Inf | accelerometer_link_3 | free | acceleration $(m/s^2)$ |
| 27/28/29 | 3-axis linear velocity of the third link | -Inf | Inf | velocimeter_link_3 | free | velocity $(m/s)$ |
| 30/31/32 | 3-axis angular velocity velocity of the third link | -Inf | Inf | gyro_link_3 | free | angular velocity $(rad/s)$ |
| 33/34/35 | 3D magnetic flux vector at the third link | -Inf | Inf | magnetometer_link_3 | free | magnetic flux density $(T)$ |
| 36/37/38 | 3-axis linear acceleration of the fourth link (including gravity) | -Inf | Inf | accelerometer_link_4 | free | acceleration $(m/s^2)$ |
| 39/40/41 | 3-axis linear velocity of the fourth link | -Inf | Inf | velocimeter_link_4 | free | velocity $(m/s)$ |
| 42/43/44 | 3-axis angular velocity velocity of the fourth link | -Inf | Inf | gyro_link_4 | free | angular velocity $(rad/s)$ |
| 45/46/47 | 3D magnetic flux vector at the fourth link | -Inf | Inf | magnetometer_link_4 | free | magnetic flux density $(T)$ |
| 48/49/50 | 3-axis linear acceleration of the fifth link (including gravity) | -Inf | Inf | accelerometer_link_5 | free | acceleration $(m/s^2)$ |
| 51/52/53 | 3-axis linear velocity of the fifth link | -Inf | Inf | velocimeter_link_5 | free | velocity $(m/s)$ |
| 54/55/56 | 3-axis angular velocity velocity of the fifth link | -Inf | Inf | gyro_link_5 | free | angular velocity $(rad/s)$ |
| 57/58/59 | 3D magnetic flux vector at the fifth link | -Inf | Inf | magnetometer_link_5 | free | magnetic flux density $(T)$ |
| 60/61/62 | 3-axis linear acceleration of the sixth link (including gravity) | -Inf | Inf | accelerometer_link_6 | free | acceleration $(m/s^2)$ |
| 63/64/65 | 3-axis linear velocity of the sixth link | -Inf | Inf | velocimeter_link_6 | free | velocity $(m/s)$ |
| 66/67/68 | 3-axis angular velocity velocity of the sixth link | -Inf | Inf | gyro_link_6 | free | angular velocity $(rad/s)$ |
| 69/70/71 | 3D magnetic flux vector at the sixth link | -Inf | Inf | magnetometer_link_6 | free | magnetic flux density $(T)$ |
| 72/73/74 | 3-axis linear acceleration of the seventh link (including gravity) | -Inf | Inf | accelerometer_link_7 | free | acceleration $(m/s^2)$ |
| 75/76/77 | 3-axis linear velocity of the seventh link | -Inf | Inf | velocimeter_link_7 | free | velocity $(m/s)$ |
| 78/79/80 | 3-axis angular velocity velocity of the seventh link | -Inf | Inf | gyro_link_7 | free | angular velocity $(rad/s)$ |
| 81/82/83 | 3D magnetic flux vector at the seventh link | -Inf | Inf | magnetometer_link_7 | free | magnetic flux density $(T)$ |
| 84 | Angle of the first joint | -Inf | Inf | jointpos_joint_1 | hinge | angle $(rad)$ |
| 85 | Angle of the second joint | -Inf | Inf | jointpos_joint_2 | hinge | angle $(rad)$ |
| 86 | Angle of the third joint | -Inf | Inf | jointpos_joint_3 | hinge | angle $(rad)$ |
| 87 | Angle of the fourth joint | -Inf | Inf | jointpos_joint_4 | hinge | angle $(rad)$ |
| 88 | Angle of the fifth joint | -Inf | Inf | jointpos_joint_5 | hinge | angle $(rad)$ |
| 89 | Angle of the sixth joint | -Inf | Inf | jointpos_joint_6 | hinge | angle $(rad)$ |
| 90 | Angular velocity of the first joint | -Inf | Inf | jointvel_joint_1 | hinge | angular velocity $(rad/s)$ |
| 91 | Angular velocity of the second joint | -Inf | Inf | jointvel_joint_2 | hinge | angular velocity $(rad/s)$ |
| 92 | Angular velocity of the third joint | -Inf | Inf | jointvel_joint_3 | hinge | angular velocity $(rad/s)$ |
| 93 | Angular velocity of the fourth joint | -Inf | Inf | jointvel_joint_4 | hinge | angular velocity $(rad/s)$ |
| 94 | Angular velocity of the fifth joint | -Inf | Inf | jointvel_joint_5 | hinge | angular velocity $(rad/s)$ |
| 95 | Angular velocity of the sixth joint | -Inf | Inf | jointvel_joint_6 | hinge | angular velocity $(rad/s)$ |
| 96 | Contact force at the end effector | 0 | Inf | touch_end_effector | - | force$(N)$ |

Table 15: Action space of Drone

| Num | Action | Min | Max | Name in XML | Joint | Unit |
|---|---|---|---|---|---|---|
| 0 | Extra thrust force applied on the first propeller | -1 | 1 | - | - | force ($N$) |
| 1 | Extra thrust force applied on the second propeller | -1 | 1 | - | - | force ($N$) |
| 2 | Extra thrust force applied on the third propeller | -1 | 1 | - | - | force ($N$) |
| 3 | Extra thrust force applied on the fourth propeller | -1 | 1 | - | - | force ($N$) |

Table 16: Observation space of Drone

| Num | Observation | Min | Max | Name in XML | Joint | Unit |
|---|---|---|---|---|---|---|
| 0/1/2 | 3-axis linear acceleration of the torso (including gravity) | -Inf | Inf | accelerometer | free | acceleration ($m/s^2$) |
| 3/4/5 | 3-axis linear velocity of the torso | -Inf | Inf | velocimeter | free | velocity ($m/s$) |
| 6/7/8 | 3-axis angular velocity velocity of the torso | -Inf | Inf | gyro | free | angular velocity ($rad/s$) |
| 9/10/11 | 3D magnetic flux vector at the torso | -Inf | Inf | magnetometer | free | magnetic flux density ($T$) |
| 12 | Contact force at the upper point of the first propeller | 0 | Inf | touch_p1a | - | force($N$) |
| 13 | Contact force at the lower point of the first propeller | 0 | Inf | touch_p1b | - | force($N$) |
| 14 | Contact force at the upper point of the second propeller | 0 | Inf | touch_p2a | - | force($N$) |
| 15 | Contact force at the lower point of the second propeller | 0 | Inf | touch_p2b | - | force($N$) |
| 16 | Contact force at the upper point of the third propeller | 0 | Inf | touch_p3a | - | force($N$) |
| 17 | Contact force at the lower point of the third propeller | 0 | Inf | touch_p3b | - | force($N$) |
| 18 | Contact force at the upper point of the fourth propeller | 0 | Inf | touch_p4a | - | force($N$) |
| 19 | Contact force at the lower point of the fourth propeller | 0 | Inf | touch_p4b | - | force($N$) |

## A.2 Observation Space Options and Desiderata

The observation spaces are also updated to match the new 3D tasks. The 3D compasses and 3D pseudo-lidars are introduced for 3D robots to sensor the position of targets in 3D space. Different from the single lidar system of the original environmet, the Advanced Safety Gym allows to apply multiple lidars on different parts of the robot. For example, in Figure 5a the Arm robot is equipped with a 3D lidar and a 3D compass on each joint to obtain more environment information. Figure 5b shows a drone equipped with two 3D lidars to observe the 3D hazards and the 3D goal. The "lidar halos" of two lidars are distributed on two sphere with different radius. The number of "lidar halos" is configurable for more dense observations.

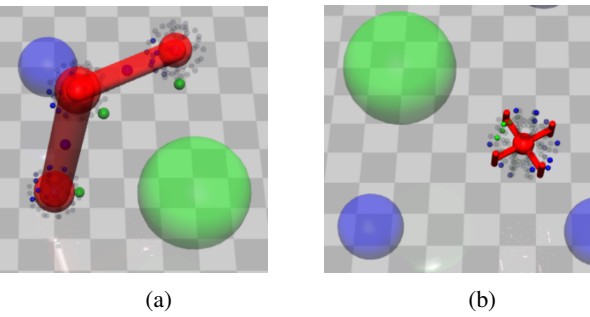

(a)          (b)

Figure 5: Visualizations of observation spaces

## A.3 Layout Randomization Options and Desiderata

The layout randomization is inherited from the original Safety Gym. In order to generate 3D objects, the $z$ coordinate can be configured or randomly picked after the $x$ and $y$ coordinates are generated.

## A.4 Task and Constraint Details

Table 17: Comparison between different tasks

|  | GUARD Tasks | | | | SafetyGym Tasks | | |
|---|---|---|---|---|---|---|---|
|  | Goal | Push | Chase | Defense | Goal | Button | Push |
| Interactive task |  | ✓ | ✓ | ✓ |  |  | ✓ |
| Non-interactive task | ✓ |  |  |  | ✓ | ✓ |  |
| Contact task |  | ✓ | ✓ | ✓ |  | ✓ | ✓ |
| Non-contact task | ✓ |  | ✓ | ✓ | ✓ |  |  |
| 2D task | ✓ | ✓ | ✓ | ✓ | ✓ | ✓ | ✓ |
| 3D task | ✓ |  | ✓ | ✓ |  |  |  |
| Movable target |  | ✓ | ✓ | ✓ |  |  | ✓ |
| Immovable target | ✓ |  |  |  | ✓ | ✓ |  |
| Single target | ✓ | ✓ | ✓ | ✓ | ✓ | ✓ | ✓ |
| Multiple targets |  |  | ✓ | ✓ |  |  |  |
| General contact target |  | ✓ | ✓ | ✓ |  | ✓ |  |

Table 18: Comparison between different constraints

| | New Constraints | | | Inherited Constraints | | | | |
|---|---|---|---|---|---|---|---|---|
| | Ghosts | Ghosts 3D | Hazards 3D | Hazards | Vases | Pillars | Buttons | Gremlins |
| Trespassable | ✓ | ✓ | | | ✓ | ✓ | ✓ | ✓ |
| Untrespassable | ✓ | ✓ | ✓ | ✓ | | | | |
| Immovable | | | ✓ | ✓ | | ✓ | ✓ | |
| Passively movable | | | | | ✓ | | | |
| Actively movable | ✓ | ✓ | | | | | | ✓ |
| 3D motion | | ✓ | ✓ | | | | | |
| 2D motion | ✓ | ✓ | ✓ | ✓ | ✓ | ✓ | ✓ | ✓ |

## A.5 Dynamics of movable objects

We begin by defining the distance vector $d_{\text{origin}} = x_{\text{origin}} - x_{\text{object}}$, which represents the distance from the position of the dynamic object $x_{\text{object}}$ to the origin point of the world framework $x_{\text{origin}}$. By default, the origin point is set to $(0, 0, 0)$. Next, we define the distance vector $d_{\text{robot}} = x_{\text{robot}} - x_{\text{object}}$, which represents the distance from the dynamic object $x_{\text{object}}$ to the position of the robot $x_{\text{robot}}$. We introduce two parameters: $r_0$, which defines a circular area centered at the origin point within which the objects are limited to move. $r_1$, which represents the threshold distance that the dynamic objects strive to maintain from the robot. Finally, we have three configurable non-negative velocity constants for the dynamic objects: $v_0$, $v_1$, and $v_2$.

### A.5.1 Dynamics of targets of Chase task

$$
\dot{x}_{object} = \begin{cases} v_0 * d_{origin}, & \text{if } \|d_{origin}\| > r_0 \\ -v_1 * d_{robot}, & \text{if } \|d_{origin}\| \leq r_0 \text{ and } \|d_{robot}\| \leq r_1 \\ 0, & \text{if } \|d_{origin}\| \leq r_0 \text{ and } \|d_{robot}\| > r_1 \end{cases} , \tag{8}
$$

### A.5.2 Dynamics of targets of Defense task

$$
\dot{x}_{object} = \begin{cases} v_0 * d_{origin}, & \text{if } \|d_{origin}\| > r_0 \\ -v_1 * d_{robot}, & \text{if } \|d_{origin}\| \leq r_0 \text{ and } \|d_{robot}\| \leq r_1 \\ v_2 * d_{origin}, & \text{if } \|d_{origin}\| \leq r_0 \text{ and } \|d_{robot}\| > r_1 \end{cases} , \tag{9}
$$

### A.5.3 Dynamics of ghost and 3D ghost

$$
\dot{x}_{object} = \begin{cases} v_0 * d_{origin}, & \text{if } \|d_{origin}\| > r_0 \\ v_1 * d_{robot}, & \text{if } \|d_{origin}\| \leq r_0 \text{ and } \|d_{robot}\| > r_1 \\ 0, & \text{if } \|d_{origin}\| \leq r_0 \text{ and } \|d_{robot}\| \leq r_1 \end{cases} , \tag{10}
$$

## B  Expeiment Details

The full GUARD codebase is available online at

> https://github.com/intelligent-control-lab/guard

The GUARD implementation is partially inspired by Safety Gym [Ray et al., 2019] and Spinningup [Achiam, 2018] which are both under MIT license.

### B.1  Policy Settings

The hyper-parameters used in our experiments are listed in Table 19 as default.

Our experiments use separate multi-layer perception with $tanh$ activations for the policy network, value network and cost network. Each network consists of two hidden layers of size (64,64). All of the networks are trained using $Adam$ optimizer with learning rate of 0.01.

We apply an on-policy framework in our experiments. During each epoch the agent interact $B$ times with the environment and then perform a policy update based on the experience collected from the current epoch. The maximum length of the trajectory is set to 1000 and the total epoch number $N$ is set to 200 as default. In our experiments the Walker and the Ant were trained for 1000 epochs due to the high dimension.

The policy update step is based on the scheme of TRPO, which performs up to 100 steps of back-tracking with a coefficient of 0.8 for line searching.

For all experiments, we use a discount factor of $\gamma = 0.99$, an advantage discount factor $\lambda = 0.95$, and a KL-divergence step size of $\delta_{KL} = 0.02$.

For experiments which consider cost constraints we adopt a target cost $\delta_c = 0.0$ to pursue a zero-violation policy.

Other unique hyper-parameters for each algorithms are hand-tuned to attain reasonable performance.

Each model is trained on a server with a 48-core Intel(R) Xeon(R) Silver 4214 CPU @ 2.2.GHz, Nvidia RTX A4000 GPU with 16GB memory, and Ubuntu 20.04.

For low-dimensional tasks, we train each model for 6e6 steps which takes around seven hours. For high-dimensional tasks, we train each model for 3e7 steps which takes around 60 hours.

### B.2  Experiment tasks

### B.3  Metrics Comparison

we report all the 72 results of our test suites by three metrics:

- The average episode return $J_r$.
- The average episodic sum of costs $M_c$.
- The average cost over the entirety of training $\rho_c$.

All of the three metrics were obtained from the final epoch after convergence. Each metric was averaged over two random seed.

Table 19: Important hyper-parameters of different algorithms in our experiments

| Policy Parameter | | TRPO | TRPO-Lagrangian | TRPO-SL [18' Dalal] | TRPO-USL | TRPO-IPO | TRPO-FAC | CPO | PCPO |
|---|---|---|---|---|---|---|---|---|---|
| Epochs | $N$ | 200 | 200 | 200 | 200 | 200 | 200 | 200 | 200 |
| Steps per epoch | $B$ | 30000 | 30000 | 30000 | 30000 | 30000 | 30000 | 30000 | 30000 |
| Maximum length of trajectory | $L$ | 1000 | 1000 | 1000 | 1000 | 1000 | 1000 | 1000 | 1000 |
| Policy network hidden layers | | (64, 64) | (64, 64) | (64, 64) | (64, 64) | (64, 64) | (64, 64) | (64, 64) | (64, 64) |
| Discount factor | $\gamma$ | 0.99 | 0.99 | 0.99 | 0.99 | 0.99 | 0.99 | 0.99 | 0.99 |
| Advantage discount factor | $\lambda$ | 0.97 | 0.97 | 0.97 | 0.97 | 0.97 | 0.97 | 0.97 | 0.97 |
| TRPO backtracking steps | | 100 | 100 | 100 | 100 | 100 | 100 | 100 | - |
| TRPO backtracking coefficient | | 0.8 | 0.8 | 0.8 | 0.8 | 0.8 | 0.8 | 0.8 | - |
| Target KL | $\delta_{KL}$ | 0.02 | 0.02 | 0.02 | 0.02 | 0.02 | 0.02 | 0.02 | 0.02 |
| Value network hidden layers | | (64, 64) | (64, 64) | (64, 64) | (64, 64) | (64, 64) | (64, 64) | (64, 64) | (64, 64) |
| Value network iteration | | 80 | 80 | 80 | 80 | 80 | 80 | 80 | 80 |
| Value network optimizer | | Adam | Adam | Adam | Adam | Adam | Adam | Adam | Adam |
| Value learning rate | | 0.001 | 0.001 | 0.001 | 0.001 | 0.001 | 0.001 | 0.001 | 0.001 |
| Cost network hidden layers | | - | (64, 64) | (64, 64) | (64, 64) | - | (64, 64) | (64, 64) | (64, 64) |
| Cost network iteration | | - | 80 | 80 | 80 | - | 80 | 80 | 80 |
| Cost network optimizer | | - | Adam | Adam | Adam | - | Adam | Adam | Adam |
| Cost learning rate | | - | 0.001 | 0.001 | 0.001 | - | 0.001 | 0.001 | 0.001 |
| Target Cost | $\delta_c$ | - | 0.0 | 0.0 | 0.0 | 0.0 | 0.0 | 0.0 | 0.0 |
| Lagrangian optimizer | | - | - | - | - | - | Adam | - | - |
| Lagrangian learning rate | | - | 0.005 | - | - | - | 0.0001 | - | - |
| USL correction iteration | | - | - | - | 20 | - | - | - | - |
| USL correction rate | | - | - | - | 0.05 | - | - | - | - |
| Warmup ratio | | - | - | 1/3 | 1/3 | - | - | - | - |
| IPO parameter | $t$ | - | - | - | - | 0.01 | - | - | - |
| Cost reduction | | - | - | - | - | - | - | 0.0 | - |

| Goal_Point_8Hazards |
|---|
| Goal_Point_8Ghosts |
| Goal_Swimmer_8Hazards |
| Goal_Swimmer_8Ghosts |
| Goal_Ant_8Hazards |
| Goal_Ant_8Ghosts |
| Goal_Walker_8Hazards |
| Goal_Walker_8Ghosts |
| Goal_Humanoid_8Hazards |
| Goal_Humanoid_8Ghosts |
| Goal_Hopper_8Hazards |
| Goal_Hopper_8Ghosts |
| Goal_Arm3_8Hazards |
| Goal_Arm3_8Ghosts |
| Goal_Arm6_8Hazards |
| Goal_Arm6_8Ghosts |
| Goal_Drone_8Hazards |
| Goal_Drone_8Ghosts |

(a) Goal

| Push_Point_8Hazards |
|---|
| Push_Point_8Ghosts |
| Push_Swimmer_8Hazards |
| Push_Swimmer_8Ghosts |
| Push_Ant_8Hazards |
| Push_Ant_8Ghosts |
| Push_Walker_8Hazards |
| Push_Walker_8Ghosts |
| Push_Humanoid_8Hazards |
| Push_Humanoid_8Ghosts |
| Push_Hopper_8Hazards |
| Push_Hopper_8Ghosts |
| Push_Arm3_8Hazards |
| Push_Arm3_8Ghosts |
| Push_Arm6_8Hazards |
| Push_Arm6_8Ghosts |
| Push_Drone_8Hazards |
| Push_Drone_8Ghosts |

(b) Push

| Chase_Point_8Hazards |
|---|
| Chase_Point_8Ghosts |
| Chase_Swimmer_8Hazards |
| Chase_Swimmer_8Ghosts |
| Chase_Ant_8Hazards |
| Chase_Ant_8Ghosts |
| Chase_Walker_8Hazards |
| Chase_Walker_8Ghosts |
| Chase_Humanoid_8Hazards |
| Chase_Humanoid_8Ghosts |
| Chase_Hopper_8Hazards |
| Chase_Hopper_8Ghosts |
| Chase_Arm3_8Hazards |
| Chase_Arm3_8Ghosts |
| Chase_Arm6_8Hazards |
| Chase_Arm6_8Ghosts |
| Chase_Drone_8Hazards |
| Chase_Drone_8Ghosts |

(c) Chase

| Defense_Point_8Hazards |
|---|
| Defense_Point_8Ghosts |
| Defense_Swimmer_8Hazards |
| Defense_Swimmer_8Ghosts |
| Defense_Ant_8Hazards |
| Defense_Ant_8Ghosts |
| Defense_Walker_8Hazards |
| Defense_Walker_8Ghosts |
| Defense_Humanoid_8Hazards |
| Defense_Humanoid_8Ghosts |
| Defense_Hopper_8Hazards |
| Defense_Hopper_8Ghosts |
| Defense_Arm3_8Hazards |
| Defense_Arm3_8Ghosts |
| Defense_Arm6_8Hazards |
| Defense_Arm6_8Ghosts |
| Defense_Drone_8Hazards |
| Defense_Drone_8Ghosts |

(d) Defense

Table 20: Tasks of our environments

Table 21: Metrics of nine **Goal_{Robot}_8Hazards** environments obtained from the final epoch.

Goal_Point_8Hazards

| Algorithm | $\bar{J}_r$ | $\bar{M}_c$ | $\bar{\rho}_c$ |
|---|---|---|---|
| TRPO | 26.2296 | 7.4550 | 0.0067 |
| TRPO-Lagrangian | 25.4503 | 2.5031 | **0.0034** |
| TRPO-SL | 19.0765 | 3.5200 | 0.0056 |
| TRPO-USL | 24.6524 | 7.0004 | 0.0060 |
| TRPO-IPO | 20.3057 | 4.4037 | 0.0049 |
| TRPO-FAC | **26.9707** | **2.1581** | 0.0038 |
| CPO | 25.9157 | 3.2388 | 0.0036 |
| PCPO | 24.9032 | 3.7118 | 0.0048 |

Goal_Swimmer_8Hazards

| Algorithm | $\bar{J}_r$ | $\bar{M}_c$ | $\bar{\rho}_c$ |
|---|---|---|---|
| TRPO | **31.5282** | 11.4067 | 0.0117 |
| TRPO-Lagrangian | 19.5685 | **4.3231** | **0.0074** |
| TRPO-SL | 9.2362 | 4.4453 | 0.0075 |
| TRPO-USL | 30.2756 | 10.2352 | 0.0100 |
| TRPO-IPO | 9.5714 | 7.9993 | 0.0079 |
| TRPO-FAC | 24.8486 | 7.8014 | 0.0085 |
| CPO | 26.6166 | 9.2452 | 0.0095 |
| PCPO | 24.4054 | 9.3452 | 0.0094 |

Goal_Ant_8Hazards

| Algorithm | $\bar{J}_r$ | $\bar{M}_c$ | $\bar{\rho}_c$ |
|---|---|---|---|
| TRPO | 59.3694 | 7.9737 | 0.0097 |
| TRPO-Lagrangian | 35.0180 | **2.7954** | **0.0056** |
| TRPO-SL | 24.0752 | 45.9755 | 0.0355 |
| TRPO-USL | 59.2213 | 9.2237 | 0.0096 |
| TRPO-IPO | 2.6040 | 6.3006 | 0.0059 |
| TRPO-FAC | 48.2685 | 5.6736 | 0.0071 |
| CPO | 60.2093 | 8.1194 | 0.0092 |
| PCPO | **60.3654** | 8.9137 | 0.0091 |

Goal_Walker_8Hazards

| Algorithm | $\bar{J}_r$ | $\bar{M}_c$ | $\bar{\rho}_c$ |
|---|---|---|---|
| TRPO | 56.7139 | 9.8112 | 0.0104 |
| TRPO-Lagrangian | 33.7839 | **3.3714** | **0.0053** |
| TRPO-SL | 39.9848 | 12.7370 | 0.0128 |
| TRPO-USL | **57.1097** | 9.9469 | 0.0097 |
| TRPO-IPO | 7.2728 | 6.7115 | 0.0068 |
| TRPO-FAC | 42.6250 | 4.4426 | 0.0062 |
| CPO | 51.9246 | 8.0409 | 0.0082 |
| PCPO | 55.0100 | 10.0377 | 0.0089 |

Goal_Humanoid_8Hazards

| Algorithm | $\bar{J}_r$ | $\bar{M}_c$ | $\bar{\rho}_c$ |
|---|---|---|---|
| TRPO | 11.6758 | 8.2332 | 0.0079 |
| TRPO-Lagrangian | 6.1294 | 7.6847 | **0.0066** |
| TRPO-SL | 9.1517 | 10.1473 | 0.0091 |
| TRPO-USL | 10.9310 | 9.2950 | 0.0079 |
| TRPO-IPO | 2.5561 | 9.0792 | 0.0071 |
| TRPO-FAC | 10.0730 | 8.3481 | 0.0068 |
| CPO | **11.9573** | **6.0618** | 0.0074 |
| PCPO | 11.6731 | 6.8256 | 0.0074 |

Goal_Hopper_8Hazards

| Algorithm | $\bar{J}_r$ | $\bar{M}_c$ | $\bar{\rho}_c$ |
|---|---|---|---|
| TRPO | **32.8406** | 7.3477 | 0.0082 |
| TRPO-Lagrangian | 24.2180 | 6.4342 | **0.0069** |
| TRPO-SL | 26.1236 | 8.9366 | 0.0098 |
| TRPO-USL | 32.5692 | 8.1526 | 0.0080 |
| TRPO-IPO | 4.0118 | 7.2667 | 0.0082 |
| TRPO-FAC | 28.1388 | **6.3430** | 0.0076 |
| CPO | 27.2544 | 8.0783 | 0.0076 |
| PCPO | 30.7637 | 6.4343 | 0.0076 |

Goal_Arm3_8Hazards

| Algorithm | $\bar{J}_r$ | $\bar{M}_c$ | $\bar{\rho}_c$ |
|---|---|---|---|
| TRPO | 19.8716 | 23.8574 | 0.0293 |
| TRPO-Lagrangian | 6.0512 | 2.1411 | **0.0057** |
| TRPO-SL | 4.2161 | **0.4820** | 0.0115 |
| TRPO-USL | 15.6522 | 8.6754 | 0.0163 |
| TRPO-IPO | 2.4211 | 12.5567 | 0.0199 |
| TRPO-FAC | 10.0948 | 3.3072 | 0.0085 |
| CPO | 16.2682 | 22.1031 | 0.0210 |
| PCPO | **21.5110** | 16.2963 | 0.0211 |

Goal_Arm6_8Hazards

| Algorithm | $\bar{J}_r$ | $\bar{M}_c$ | $\bar{\rho}_c$ |
|---|---|---|---|
| TRPO | 4.3703 | 15.0087 | 0.0206 |
| TRPO-Lagrangian | 1.2386 | 6.8767 | **0.0107** |
| TRPO-SL | 2.1136 | 14.1806 | 0.0136 |
| TRPO-USL | 2.5704 | 9.4493 | 0.0186 |
| TRPO-IPO | 0.8242 | **5.5569** | 0.0129 |
| TRPO-FAC | 2.4243 | 8.9828 | 0.0124 |
| CPO | **4.3885** | 13.0115 | 0.0171 |
| PCPO | 1.1528 | 13.8961 | 0.0141 |

Goal_Drone_8Hazards

| Algorithm | $\bar{J}_r$ | $\bar{M}_c$ | $\bar{\rho}_c$ |
|---|---|---|---|
| TRPO | **19.6492** | 1.6839 | 0.0012 |
| TRPO-Lagrangian | 17.5182 | 1.0479 | 0.0010 |
| TRPO-SL | 11.0012 | **0.2030** | 0.0004 |
| TRPO-USL | 17.3535 | 1.1217 | 0.0008 |
| TRPO-IPO | 15.7189 | 0.8852 | 0.0007 |
| TRPO-FAC | 17.0156 | 1.0926 | 0.0005 |
| CPO | 18.3672 | 1.0204 | 0.0010 |
| PCPO | 5.0076 | 0.2334 | **0.0003** |

Table 22: Metrics of nine **Goal_{Robot}_8Ghosts** environments obtained from the final epoch.

Goal_Point_8Ghosts

| Algorithm | $\bar{J}_r$ | $\bar{M}_c$ | $\bar{\rho}_c$ |
|---|---|---|---|
| TRPO | 26.0478 | 6.8329 | 0.0073 |
| TRPO-Lagrangian | 26.3260 | **2.1498** | 0.0034 |
| TRPO-SL | 16.6548 | 4.0515 | 0.0058 |
| TRPO-USL | 22.1795 | 5.8895 | 0.0059 |
| TRPO-IPO | 20.1808 | 4.1169 | 0.0050 |
| TRPO-FAC | 25.9489 | 2.5654 | 0.0036 |
| CPO | **26.5064** | 2.6248 | **0.0034** |
| PCPO | 25.9672 | 3.8589 | 0.0054 |

Goal_Swimmer_8Ghosts

| Algorithm | $\bar{J}_r$ | $\bar{M}_c$ | $\bar{\rho}_c$ |
|---|---|---|---|
| TRPO | **30.3401** | 13.5808 | 0.0119 |
| TRPO-Lagrangian | 15.9952 | **2.1046** | **0.0061** |
| TRPO-SL | 7.8773 | 7.6875 | 0.0079 |
| TRPO-USL | 30.1229 | 8.9488 | 0.0105 |
| TRPO-IPO | 9.8646 | 10.0275 | 0.0091 |
| TRPO-FAC | 18.9950 | 4.4988 | 0.0069 |
| CPO | 26.6953 | 9.5202 | 0.0092 |
| PCPO | 26.2737 | 10.2204 | 0.0101 |

Goal_Ant_8Ghosts

| Algorithm | $\bar{J}_r$ | $\bar{M}_c$ | $\bar{\rho}_c$ |
|---|---|---|---|
| TRPO | 59.6760 | 10.3785 | 0.0099 |
| TRPO-Lagrangian | 28.5846 | **2.9654** | **0.0060** |
| TRPO-SL | 30.7285 | 41.2262 | 0.0342 |
| TRPO-USL | **61.2725** | 8.9165 | 0.0097 |
| TRPO-IPO | 2.9659 | 8.0972 | 0.0064 |
| TRPO-FAC | 44.2423 | 5.6508 | 0.0074 |
| CPO | 56.3422 | 9.8690 | 0.0095 |
| PCPO | 58.4684 | 9.8173 | 0.0095 |

Goal_Walker_8Ghosts

| Algorithm | $\bar{J}_r$ | $\bar{M}_c$ | $\bar{\rho}_c$ |
|---|---|---|---|
| TRPO | **63.2017** | 9.8771 | 0.0112 |
| TRPO-Lagrangian | 33.2534 | **2.5072** | **0.0054** |
| TRPO-SL | 37.8968 | 20.3758 | 0.0147 |
| TRPO-USL | 61.4547 | 9.6043 | 0.0105 |
| TRPO-IPO | 7.4640 | 9.1178 | 0.0080 |
| TRPO-FAC | 45.0094 | 4.9375 | 0.0071 |
| CPO | 60.1257 | 9.2117 | 0.0097 |
| PCPO | 43.8760 | 9.2932 | 0.0085 |

Goal_Humanoid_8Ghosts

| Algorithm | $\bar{J}_r$ | $\bar{M}_c$ | $\bar{\rho}_c$ |
|---|---|---|---|
| TRPO | 11.1891 | 9.9692 | 0.0098 |
| TRPO-Lagrangian | 5.0070 | **6.6812** | 0.0076 |
| TRPO-SL | 8.8939 | 17.0632 | 0.0107 |
| TRPO-USL | 10.6905 | 9.6248 | 0.0095 |
| TRPO-IPO | 1.0404 | 8.4966 | **0.0073** |
| TRPO-FAC | 9.2134 | 10.0716 | 0.0084 |
| CPO | 10.0778 | 10.3074 | 0.0092 |
| PCPO | **11.5003** | 9.0205 | 0.0093 |

Goal_Hopper_8Ghosts

| Algorithm | $\bar{J}_r$ | $\bar{M}_c$ | $\bar{\rho}_c$ |
|---|---|---|---|
| TRPO | **31.6643** | 8.1599 | 0.0100 |
| TRPO-Lagrangian | 14.1699 | **4.4744** | **0.0070** |
| TRPO-SL | 21.7761 | 12.4810 | 0.0122 |
| TRPO-USL | 31.2864 | 8.4550 | 0.0097 |
| TRPO-IPO | 5.4826 | 12.0015 | 0.0082 |
| TRPO-FAC | 28.8157 | 7.5453 | 0.0087 |
| CPO | 29.0408 | 7.5681 | 0.0086 |
| PCPO | 29.0858 | 8.0181 | 0.0090 |

Goal_Arm3_8Ghosts

| Algorithm | $\bar{J}_r$ | $\bar{M}_c$ | $\bar{\rho}_c$ |
|---|---|---|---|
| TRPO | 94.6660 | 35.7460 | 0.0348 |
| TRPO-Lagrangian | 15.4898 | 7.5123 | **0.0058** |
| TRPO-SL | 18.1207 | 10.7580 | 0.0174 |
| TRPO-USL | 62.1624 | 14.0682 | 0.0223 |
| TRPO-IPO | 4.0235 | 10.5251 | 0.0160 |
| TRPO-FAC | 37.9750 | **6.9701** | 0.0073 |
| CPO | 114.8705 | 15.1904 | 0.0159 |
| PCPO | **126.4001** | 10.1913 | 0.0143 |

Goal_Arm6_8Ghosts

| Algorithm | $\bar{J}_r$ | $\bar{M}_c$ | $\bar{\rho}_c$ |
|---|---|---|---|
| TRPO | 1.0157 | 49.0135 | 0.0466 |
| TRPO-Lagrangian | 0.5470 | **8.4307** | 0.0190 |
| TRPO-SL | 0.6078 | 20.5269 | 0.0356 |
| TRPO-USL | 0.9856 | 41.7054 | 0.0427 |
| TRPO-IPO | 0.7336 | 12.4453 | 0.0233 |
| TRPO-FAC | 0.7861 | 9.4493 | 0.0170 |
| CPO | **9.9993** | 22.5031 | 0.0234 |
| PCPO | 0.8845 | 15.9718 | **0.0162** |

Goal_Drone_8Ghosts

| Algorithm | $\bar{J}_r$ | $\bar{M}_c$ | $\bar{\rho}_c$ |
|---|---|---|---|
| TRPO | 17.9484 | 1.7287 | 0.0011 |
| TRPO-Lagrangian | 18.9773 | 0.9218 | 0.0008 |
| TRPO-SL | 12.1413 | **0.2500** | 0.0004 |
| TRPO-USL | 10.7517 | 0.9741 | 0.0011 |
| TRPO-IPO | 11.5210 | 0.6817 | 0.0006 |
| TRPO-FAC | **20.1014** | 0.7630 | 0.0006 |
| CPO | 18.4723 | 1.2188 | 0.0008 |
| PCPO | 6.5276 | 0.3859 | **0.0003** |

Table 23: Metrics of nine **Push_{Robot}_8Hazards** environments obtained from the final epoch.

### Push_Point_8Hazards

| Algorithm | $\bar{J}_r$ | $\bar{M}_c$ | $\bar{\rho}_c$ |
|---|---|---|---|
| TRPO | **11.3060** | 7.2536 | 0.0084 |
| TRPO-Lagrangian | 4.1189 | **1.8268** | **0.0037** |
| TRPO-SL | 3.0553 | 6.6139 | 0.0058 |
| TRPO-USL | 9.1904 | 6.6179 | 0.0064 |
| TRPO-IPO | 1.3370 | 4.0476 | 0.0051 |
| TRPO-FAC | 6.0431 | 2.1250 | 0.0039 |
| CPO | 9.7522 | 5.6406 | 0.0066 |
| PCPO | 9.1434 | 6.5665 | 0.0066 |

### Push_Swimmer_8Hazards

| Algorithm | $\bar{J}_r$ | $\bar{M}_c$ | $\bar{\rho}_c$ |
|---|---|---|---|
| TRPO | **86.1557** | 11.9235 | 0.0102 |
| TRPO-Lagrangian | 52.0782 | **4.5645** | 0.0070 |
| TRPO-SL | 13.1869 | 7.7554 | **0.0057** |
| TRPO-USL | 64.0705 | 9.4963 | 0.0085 |
| TRPO-IPO | 6.3843 | 8.4329 | 0.0077 |
| TRPO-FAC | 48.2986 | 5.8675 | 0.0064 |
| CPO | 57.4370 | 6.9551 | 0.0072 |
| PCPO | 56.2598 | 6.1634 | 0.0076 |

### Push_Ant_8Hazards

| Algorithm | $\bar{J}_r$ | $\bar{M}_c$ | $\bar{\rho}_c$ |
|---|---|---|---|
| TRPO | **13.4378** | 9.4740 | 0.0091 |
| TRPO-Lagrangian | 1.1582 | **1.5948** | **0.0043** |
| TRPO-SL | 3.5622 | 47.7602 | 0.0217 |
| TRPO-USL | 11.2763 | 9.3930 | 0.0086 |
| TRPO-IPO | 1.1986 | 5.9120 | 0.0061 |
| TRPO-FAC | 2.5905 | 2.7927 | 0.0050 |
| CPO | 12.7081 | 7.5742 | 0.0082 |
| PCPO | 11.0161 | 8.7780 | 0.0087 |

### Push_Walker_8Hazards

| Algorithm | $\bar{J}_r$ | $\bar{M}_c$ | $\bar{\rho}_c$ |
|---|---|---|---|
| TRPO | **5.0574** | 10.8840 | 0.0089 |
| TRPO-Lagrangian | 1.5035 | **2.4237** | **0.0040** |
| TRPO-SL | 1.7263 | 17.5680 | 0.0082 |
| TRPO-USL | 2.8786 | 9.3900 | 0.0078 |
| TRPO-IPO | 0.7991 | 3.6377 | 0.0070 |
| TRPO-FAC | 1.5393 | 3.2465 | 0.0047 |
| CPO | 4.3412 | 7.8450 | 0.0075 |
| PCPO | 1.1548 | 9.2470 | 0.0075 |

### Push_Humanoid_8Hazards

| Algorithm | $\bar{J}_r$ | $\bar{M}_c$ | $\bar{\rho}_c$ |
|---|---|---|---|
| TRPO | 0.9545 | 10.6542 | 0.0096 |
| TRPO-Lagrangian | 0.7407 | 3.1758 | **0.0062** |
| TRPO-SL | 0.2992 | 9.0239 | 0.0092 |
| TRPO-USL | 0.8102 | 7.3410 | 0.0093 |
| TRPO-IPO | 0.8194 | 6.0952 | 0.0074 |
| TRPO-FAC | 0.9641 | **3.0034** | 0.0068 |
| CPO | 0.8147 | 8.6884 | 0.0080 |
| PCPO | **1.0445** | 8.1230 | 0.0084 |

### Push_Hopper_8Hazards

| Algorithm | $\bar{J}_r$ | $\bar{M}_c$ | $\bar{\rho}_c$ |
|---|---|---|---|
| TRPO | **3.6134** | 10.3693 | 0.0095 |
| TRPO-Lagrangian | 0.8384 | **2.0782** | **0.0052** |
| TRPO-SL | 1.5115 | 8.2643 | 0.0080 |
| TRPO-USL | 2.3949 | 11.2835 | 0.0088 |
| TRPO-IPO | 0.3718 | 7.4184 | 0.0083 |
| TRPO-FAC | 1.0928 | 3.8033 | 0.0069 |
| CPO | 2.3108 | 11.2012 | 0.0082 |
| PCPO | 0.9565 | 8.8373 | 0.0083 |

### Push_Arm3_8Hazards

| Algorithm | $\bar{J}_r$ | $\bar{M}_c$ | $\bar{\rho}_c$ |
|---|---|---|---|
| TRPO | 0.0438 | 37.7114 | 0.0414 |
| TRPO-Lagrangian | -194.8455 | **2.7071** | **0.0062** |
| TRPO-SL | **0.0906** | 7.3980 | 0.0176 |
| TRPO-USL | -42.2457 | 10.6065 | 0.0189 |
| TRPO-IPO | -420.0890 | 25.0669 | 0.0224 |
| TRPO-FAC | -114.8912 | 7.8944 | 0.0086 |
| CPO | 0.0249 | 11.3773 | 0.0128 |
| PCPO | -30.9294 | 10.4467 | 0.0207 |

### Push_Arm6_8Hazards

| Algorithm | $\bar{J}_r$ | $\bar{M}_c$ | $\bar{\rho}_c$ |
|---|---|---|---|
| TRPO | 1.1128 | 15.9080 | 0.0190 |
| TRPO-Lagrangian | 0.9490 | 7.1961 | 0.0110 |
| TRPO-SL | -220.2115 | 38.7175 | 0.0144 |
| TRPO-USL | -0.6530 | 16.7103 | 0.0182 |
| TRPO-IPO | 1.1291 | 8.3642 | 0.0113 |
| TRPO-FAC | 1.0648 | 9.4750 | 0.0152 |
| CPO | **1.1699** | **6.6375** | **0.0103** |
| PCPO | 1.1459 | 10.0104 | 0.0112 |

### Push_Drone_8Hazards

| Algorithm | $\bar{J}_r$ | $\bar{M}_c$ | $\bar{\rho}_c$ |
|---|---|---|---|
| TRPO | 0.9332 | 0.3324 | 0.0002 |
| TRPO-Lagrangian | 1.0967 | 0.3197 | 0.0003 |
| TRPO-SL | 1.0154 | 0.0783 | 0.0001 |
| TRPO-USL | 0.9410 | 0.0996 | 0.0001 |
| TRPO-IPO | 1.0394 | 0.4229 | 0.0002 |
| TRPO-FAC | 1.0820 | 0.2380 | 0.0002 |
| CPO | **1.1261** | 0.2409 | 0.0003 |
| PCPO | 0.9844 | **0.0049** | **0.0001** |

Table 24: Metrics of nine **Chase_{Robot}_8Hazards** environments obtained from the final epoch.

### Chase_Point_8Hazards

| Algorithm | $\bar{J}_r$ | $\bar{M}_c$ | $\bar{\rho}_c$ |
|---|---|---|---|
| TRPO | **1.3122** | 3.5553 | 0.0068 |
| TRPO-Lagrangian | 1.0879 | **2.8816** | **0.0046** |
| TRPO-SL | 0.8385 | 5.6000 | 0.0058 |
| TRPO-USL | 1.1433 | 5.7574 | 0.0080 |
| TRPO-IPO | 0.7959 | 3.8632 | 0.0061 |
| TRPO-FAC | 1.0333 | 3.0887 | 0.0053 |
| CPO | 1.2897 | 5.0677 | 0.0063 |
| PCPO | 1.0035 | 7.9018 | 0.0084 |

### Chase_Swimmer_8Hazards

| Algorithm | $\bar{J}_r$ | $\bar{M}_c$ | $\bar{\rho}_c$ |
|---|---|---|---|
| TRPO | 1.2491 | 7.0269 | 0.0100 |
| TRPO-Lagrangian | -0.2346 | **4.8860** | **0.0058** |
| TRPO-SL | 0.0518 | 9.2681 | 0.0071 |
| TRPO-USL | 1.2227 | 9.2911 | 0.0103 |
| TRPO-IPO | -1.0848 | 10.5546 | 0.0080 |
| TRPO-FAC | 0.6411 | 9.1446 | 0.0078 |
| CPO | **1.2540** | 8.1671 | 0.0082 |
| PCPO | 1.2152 | 8.2717 | 0.0090 |

### Chase_Ant_8Hazards

| Algorithm | $\bar{J}_r$ | $\bar{M}_c$ | $\bar{\rho}_c$ |
|---|---|---|---|
| TRPO | 1.3504 | 6.1101 | 0.0106 |
| TRPO-Lagrangian | -0.3563 | **2.5016** | **0.0040** |
| TRPO-SL | 0.7921 | 16.9846 | 0.0222 |
| TRPO-USL | 1.3841 | 8.0640 | 0.0096 |
| TRPO-IPO | -0.9314 | 2.5529 | 0.0048 |
| TRPO-FAC | -0.0258 | 3.5439 | 0.0048 |
| CPO | **1.4104** | 5.7863 | 0.0087 |
| PCPO | 1.3122 | 6.9139 | 0.0097 |

### Chase_Walker_8Hazards

| Algorithm | $\bar{J}_r$ | $\bar{M}_c$ | $\bar{\rho}_c$ |
|---|---|---|---|
| TRPO | 0.4890 | 7.6845 | 0.0088 |
| TRPO-Lagrangian | -0.0922 | 2.5167 | 0.0045 |
| TRPO-SL | -0.2116 | 10.7167 | 0.0094 |
| TRPO-USL | 0.4639 | 7.7035 | 0.0082 |
| TRPO-IPO | -0.8223 | **2.3954** | **0.0038** |
| TRPO-FAC | -0.0368 | 2.7105 | 0.0047 |
| CPO | **0.7406** | 10.4993 | 0.0086 |
| PCPO | 0.6347 | 8.8652 | 0.0080 |

### Chase_Humanoid_8Hazards

| Algorithm | $\bar{J}_r$ | $\bar{M}_c$ | $\bar{\rho}_c$ |
|---|---|---|---|
| TRPO | **0.2330** | 12.1455 | 0.0152 |
| TRPO-Lagrangian | -0.6855 | **3.4234** | **0.0047** |
| TRPO-SL | -0.2271 | 11.8001 | 0.0121 |
| TRPO-USL | -0.1503 | 18.6011 | 0.0149 |
| TRPO-IPO | -0.8074 | 6.4163 | 0.0054 |
| TRPO-FAC | -0.5826 | 3.6663 | 0.0050 |
| CPO | -0.3322 | 12.1665 | 0.0109 |
| PCPO | -0.0971 | 10.3441 | 0.0113 |

### Chase_Hopper_8Hazards

| Algorithm | $\bar{J}_r$ | $\bar{M}_c$ | $\bar{\rho}_c$ |
|---|---|---|---|
| TRPO | **0.6099** | 12.1675 | 0.0134 |
| TRPO-Lagrangian | -0.3641 | 3.2170 | **0.0039** |
| TRPO-SL | 0.4957 | 5.4355 | 0.0089 |
| TRPO-USL | 0.4819 | 11.0919 | 0.0123 |
| TRPO-IPO | -0.7766 | 6.1236 | 0.0061 |
| TRPO-FAC | -0.3651 | 3.7391 | 0.0055 |
| CPO | 0.4829 | 6.7117 | 0.0083 |
| PCPO | -0.1457 | 7.4290 | 0.0068 |

### Chase_Arm3_8Hazards

| Algorithm | $\bar{J}_r$ | $\bar{M}_c$ | $\bar{\rho}_c$ |
|---|---|---|---|
| TRPO | 0.7772 | 20.6230 | 0.0312 |
| TRPO-Lagrangian | -0.2739 | 5.1692 | **0.0079** |
| TRPO-SL | 0.0007 | 4.2869 | 0.0142 |
| TRPO-USL | 0.7825 | 14.1736 | 0.0284 |
| TRPO-IPO | -0.4137 | 10.6685 | 0.0223 |
| TRPO-FAC | 0.3648 | **3.3449** | 0.0127 |
| CPO | **0.8051** | 17.4917 | 0.0252 |
| PCPO | 0.7355 | 25.8202 | 0.0291 |

### Chase_Arm6_8Hazards

| Algorithm | $\bar{J}_r$ | $\bar{M}_c$ | $\bar{\rho}_c$ |
|---|---|---|---|
| TRPO | -0.3969 | 60.5704 | 0.0598 |
| TRPO-Lagrangian | -0.4860 | **2.4602** | **0.0075** |
| TRPO-SL | -0.5420 | 12.1256 | 0.0237 |
| TRPO-USL | -0.5734 | 53.4455 | 0.0575 |
| TRPO-IPO | **-0.2855** | 11.6769 | 0.0085 |
| TRPO-FAC | -0.3083 | 13.2429 | 0.0263 |
| CPO | -0.3278 | 16.9609 | 0.0247 |
| PCPO | -0.2883 | 45.6164 | 0.0463 |

### Chase_Drone_8Hazards

| Algorithm | $\bar{J}_r$ | $\bar{M}_c$ | $\bar{\rho}_c$ |
|---|---|---|---|
| TRPO | 1.0351 | 0.6939 | 0.0008 |
| TRPO-Lagrangian | 0.8211 | 1.3456 | 0.0008 |
| TRPO-SL | -1.3055 | 0.2603 | **0.0002** |
| TRPO-USL | 0.7461 | 1.2159 | 0.0006 |
| TRPO-IPO | 0.2518 | 0.5786 | 0.0005 |
| TRPO-FAC | **1.1192** | 0.2374 | 0.0006 |
| CPO | 0.7682 | 0.9075 | 0.0006 |
| PCPO | 0.6172 | 0.6374 | 0.0012 |

Table 25: Metrics of nine **Defense_{Robot}_8Hazards** environments obtained from the final epoch.

Defense_Point_8Hazards

| Algorithm | $\bar{J}_r$ | $\bar{M}_c$ | $\bar{\rho}_c$ |
|---|---|---|---|
| TRPO | **71.7851** | 37.5050 | 0.0308 |
| TRPO-Lagrangian | -12.2159 | 1.1776 | 0.0026 |
| TRPO-SL | -89.8828 | 3.1691 | 0.0070 |
| TRPO-USL | -109.7828 | 9.9285 | 0.0086 |
| TRPO-IPO | -330.4252 | **0.7309** | 0.0035 |
| TRPO-FAC | -269.0397 | 0.7334 | **0.0015** |
| CPO | 36.7643 | 7.1534 | 0.0071 |
| PCPO | 19.0943 | 1.9388 | 0.0048 |

Defense_Swimmer_8Hazards

| Algorithm | $\bar{J}_r$ | $\bar{M}_c$ | $\bar{\rho}_c$ |
|---|---|---|---|
| TRPO | 119.9896 | 44.5965 | 0.0405 |
| TRPO-Lagrangian | -85.0177 | **0.2487** | **0.0031** |
| TRPO-SL | -41.8928 | 1.3295 | 0.0118 |
| TRPO-USL | **139.8915** | 13.5482 | 0.0150 |
| TRPO-IPO | -233.1962 | 7.6313 | 0.0070 |
| TRPO-FAC | -91.7454 | 0.8809 | 0.0032 |
| CPO | 34.3226 | 2.7346 | 0.0072 |
| PCPO | 91.1387 | 5.1068 | 0.0084 |

Defense_Ant_8Hazards

| Algorithm | $\bar{J}_r$ | $\bar{M}_c$ | $\bar{\rho}_c$ |
|---|---|---|---|
| TRPO | **65.9815** | 46.1871 | 0.0214 |
| TRPO-Lagrangian | -190.9671 | 1.5799 | **0.0040** |
| TRPO-SL | -15.0035 | 14.7914 | 0.0143 |
| TRPO-USL | -9.1186 | 25.8625 | 0.0126 |
| TRPO-IPO | -205.8713 | 6.0119 | 0.0044 |
| TRPO-FAC | -204.4595 | **1.4105** | 0.0041 |
| CPO | -22.6369 | 17.9356 | 0.0132 |
| PCPO | -42.0119 | 17.1633 | 0.0120 |

Defense_Walker_8Hazards

| Algorithm | $\bar{J}_r$ | $\bar{M}_c$ | $\bar{\rho}_c$ |
|---|---|---|---|
| TRPO | **63.0381** | 52.1661 | 0.0326 |
| TRPO-Lagrangian | -221.9464 | **0.8080** | **0.0032** |
| TRPO-SL | -28.2392 | 21.2179 | 0.0142 |
| TRPO-USL | 19.2097 | 23.4844 | 0.0182 |
| TRPO-IPO | -213.4079 | 2.7606 | 0.0045 |
| TRPO-FAC | -183.6202 | 1.6905 | 0.0035 |
| CPO | 32.0705 | 14.7761 | 0.0151 |
| PCPO | 43.8441 | 17.0562 | 0.0161 |

Defense_Humanoid_8Hazards

| Algorithm | $\bar{J}_r$ | $\bar{M}_c$ | $\bar{\rho}_c$ |
|---|---|---|---|
| TRPO | -279.6928 | 4.3248 | 0.0042 |
| TRPO-Lagrangian | -287.5846 | 3.0248 | 0.0035 |
| TRPO-SL | -325.6846 | 2.0650 | 0.0039 |
| TRPO-USL | -318.2901 | 3.5935 | 0.0043 |
| TRPO-IPO | -281.2530 | 4.3968 | 0.0038 |
| TRPO-FAC | -271.4645 | **2.0044** | **0.0034** |
| CPO | **-246.6409** | 5.8980 | 0.0049 |
| PCPO | -317.0349 | 2.9953 | 0.0040 |

Defense_Hopper_8Hazards

| Algorithm | $\bar{J}_r$ | $\bar{M}_c$ | $\bar{\rho}_c$ |
|---|---|---|---|
| TRPO | -79.4386 | 26.9427 | 0.0202 |
| TRPO-Lagrangian | -304.2345 | 1.1963 | **0.0029** |
| TRPO-SL | -207.0506 | 8.9198 | 0.0138 |
| TRPO-USL | **57.7316** | 28.5037 | 0.0234 |
| TRPO-IPO | -248.0784 | 6.6735 | 0.0046 |
| TRPO-FAC | -233.1694 | **0.7496** | 0.0038 |
| CPO | -271.5419 | 8.3413 | 0.0077 |
| PCPO | -279.4999 | 7.2803 | 0.0077 |

Defense_Arm3_8Hazards

| Algorithm | $\bar{J}_r$ | $\bar{M}_c$ | $\bar{\rho}_c$ |
|---|---|---|---|
| TRPO | 169.5352 | 22.0750 | 0.0301 |
| TRPO-Lagrangian | 151.7291 | **0.7971** | **0.0056** |
| TRPO-SL | 112.3637 | 1.1085 | 0.0160 |
| TRPO-USL | 164.4992 | 5.3212 | 0.0163 |
| TRPO-IPO | 94.1636 | 9.1085 | 0.0171 |
| TRPO-FAC | **180.9871** | 1.7731 | 0.0064 |
| CPO | 167.4984 | 16.4595 | 0.0162 |
| PCPO | 160.2841 | 22.2282 | 0.0189 |

Defense_Arm6_8Hazards

| Algorithm | $\bar{J}_r$ | $\bar{M}_c$ | $\bar{\rho}_c$ |
|---|---|---|---|
| TRPO | **183.9203** | 56.5334 | 0.0548 |
| TRPO-Lagrangian | 169.9900 | **1.0108** | **0.0045** |
| TRPO-SL | 171.8430 | 13.3277 | 0.0229 |
| TRPO-USL | 183.7060 | 52.3346 | 0.0528 |
| TRPO-IPO | 127.3447 | 3.8719 | 0.0051 |
| TRPO-FAC | 175.8257 | 2.3101 | 0.0109 |
| CPO | 174.7701 | 22.8158 | 0.0346 |
| PCPO | 174.4207 | 30.1276 | 0.0264 |

Defense_Drone_8Hazards

| Algorithm | $\bar{J}_r$ | $\bar{M}_c$ | $\bar{\rho}_c$ |
|---|---|---|---|
| TRPO | -241.5720 | 0.0771 | 0.0002 |
| TRPO-Lagrangian | -245.7311 | 0.2276 | 0.0002 |
| TRPO-SL | -371.7727 | **0.0000** | **0.0001** |
| TRPO-USL | -336.7727 | 0.2161 | 0.0001 |
| TRPO-IPO | -275.5550 | 0.2600 | 0.0002 |
| TRPO-FAC | -215.4844 | 0.0691 | 0.0001 |
| CPO | **-212.1858** | 0.0236 | 0.0002 |
| PCPO | -219.4308 | 0.3358 | 0.0003 |

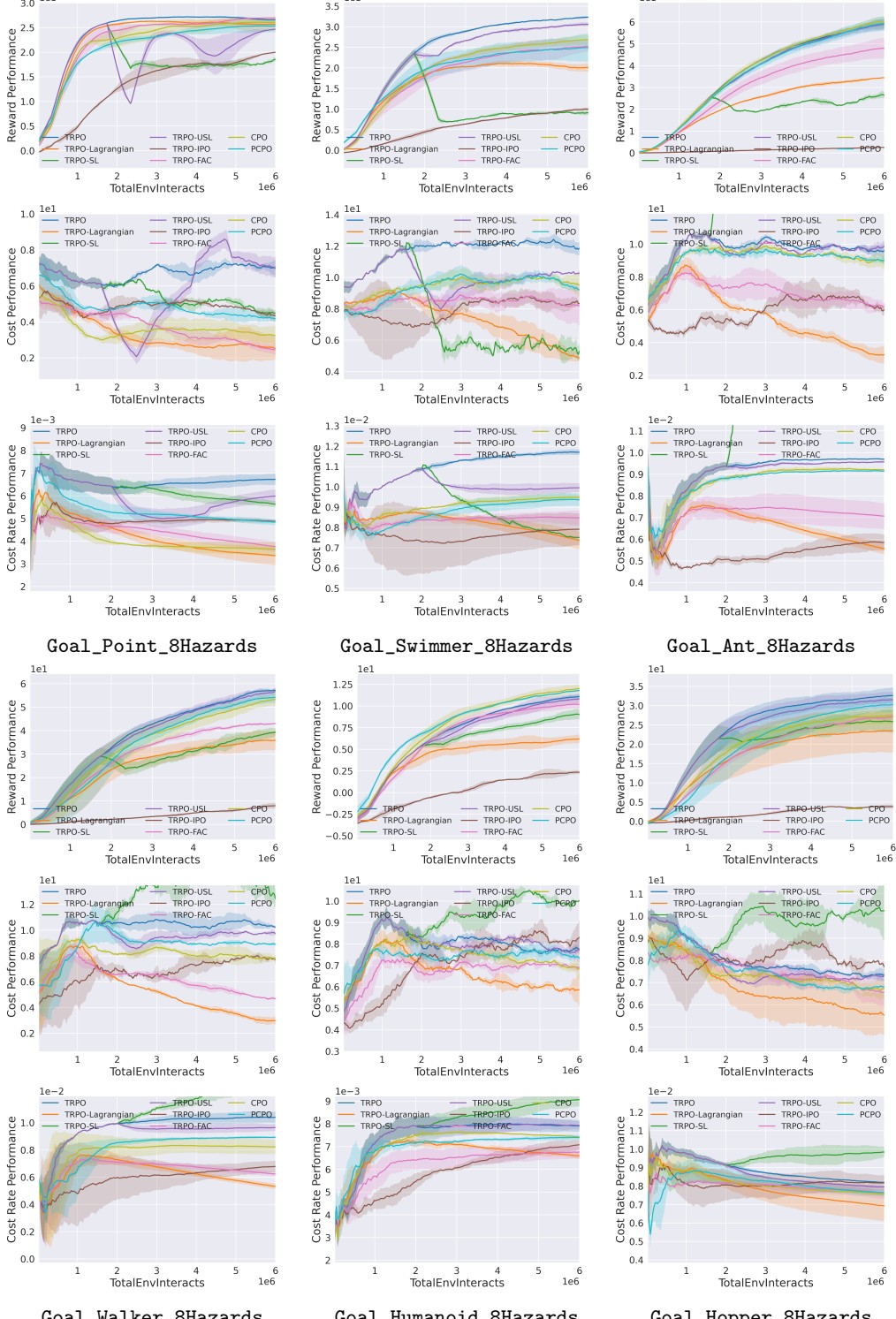

**Goal_Point_8Hazards**       **Goal_Swimmer_8Hazards**       **Goal_Ant_8Hazards**

**Goal_Walker_8Hazards**       **Goal_Humanoid_8Hazards**       **Goal_Hopper_8Hazards**

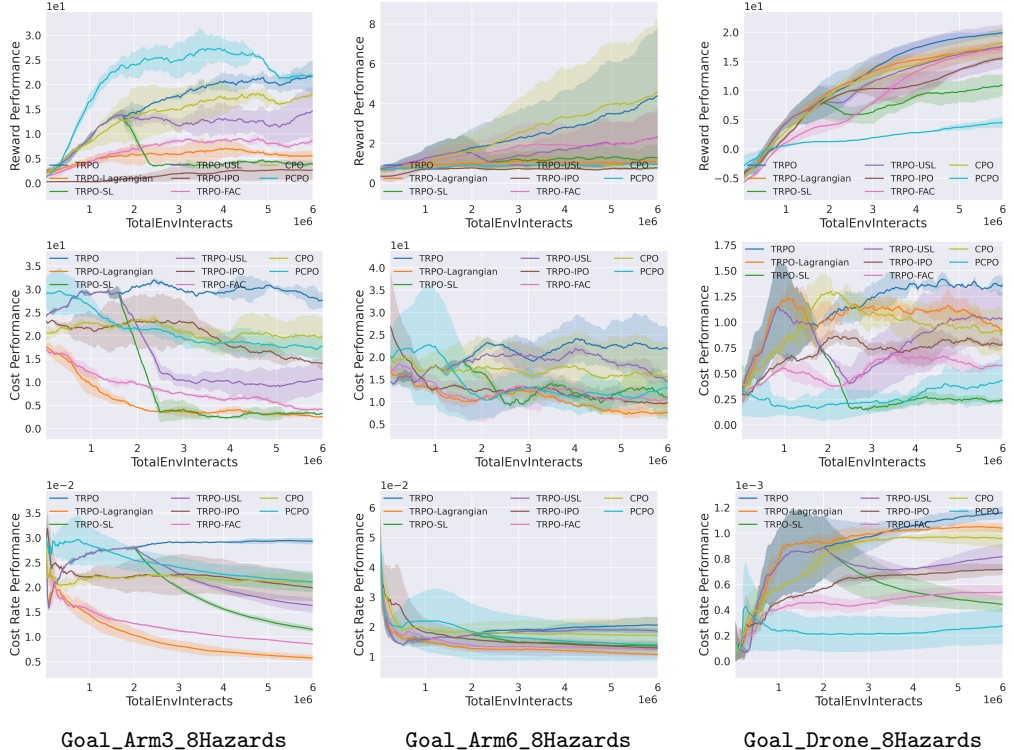

**Goal_Arm3_8Hazards**          **Goal_Arm6_8Hazards**          **Goal_Drone_8Hazards**

Figure 6: `Goal_{Robot}_8Hazards`

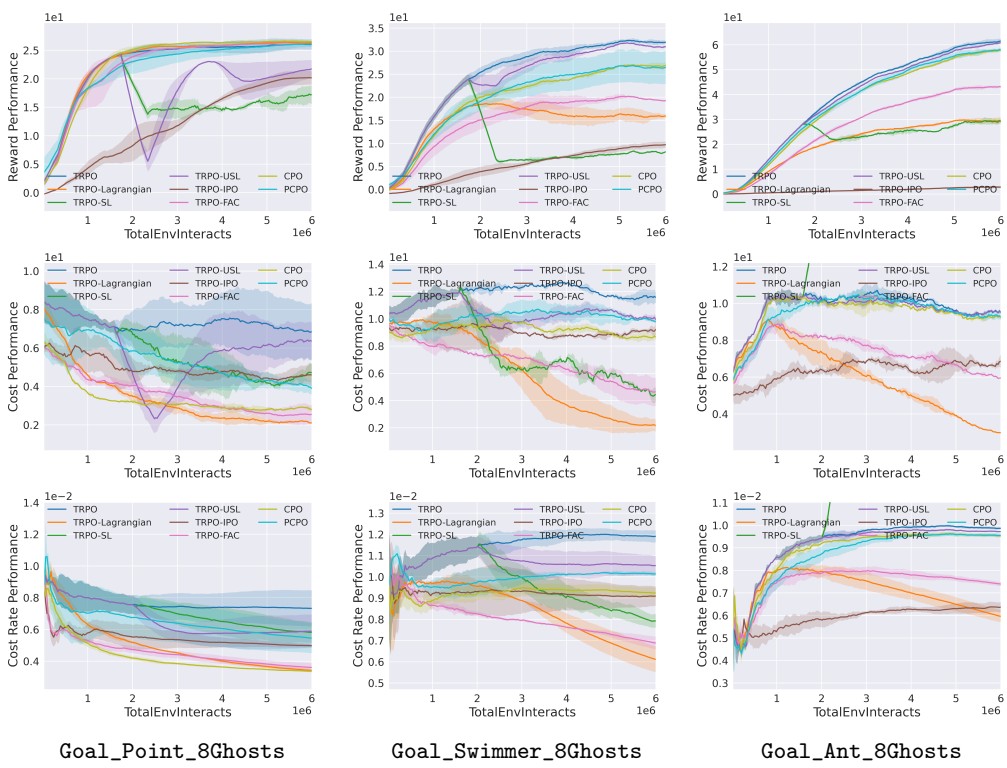

**Goal_Point_8Ghosts**          **Goal_Swimmer_8Ghosts**          **Goal_Ant_8Ghosts**

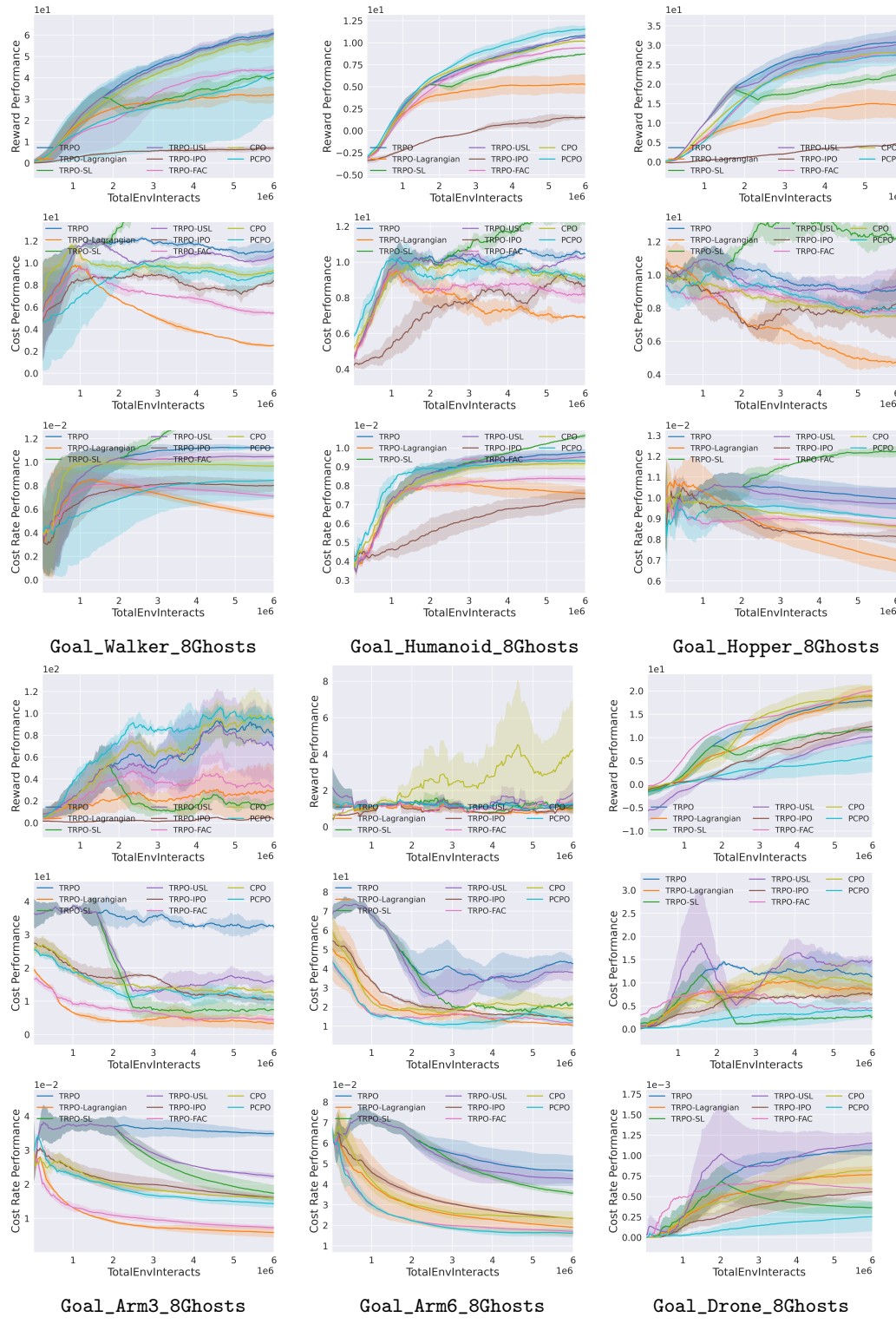

Figure 7: `Goal_{Robot}_8Ghosts`

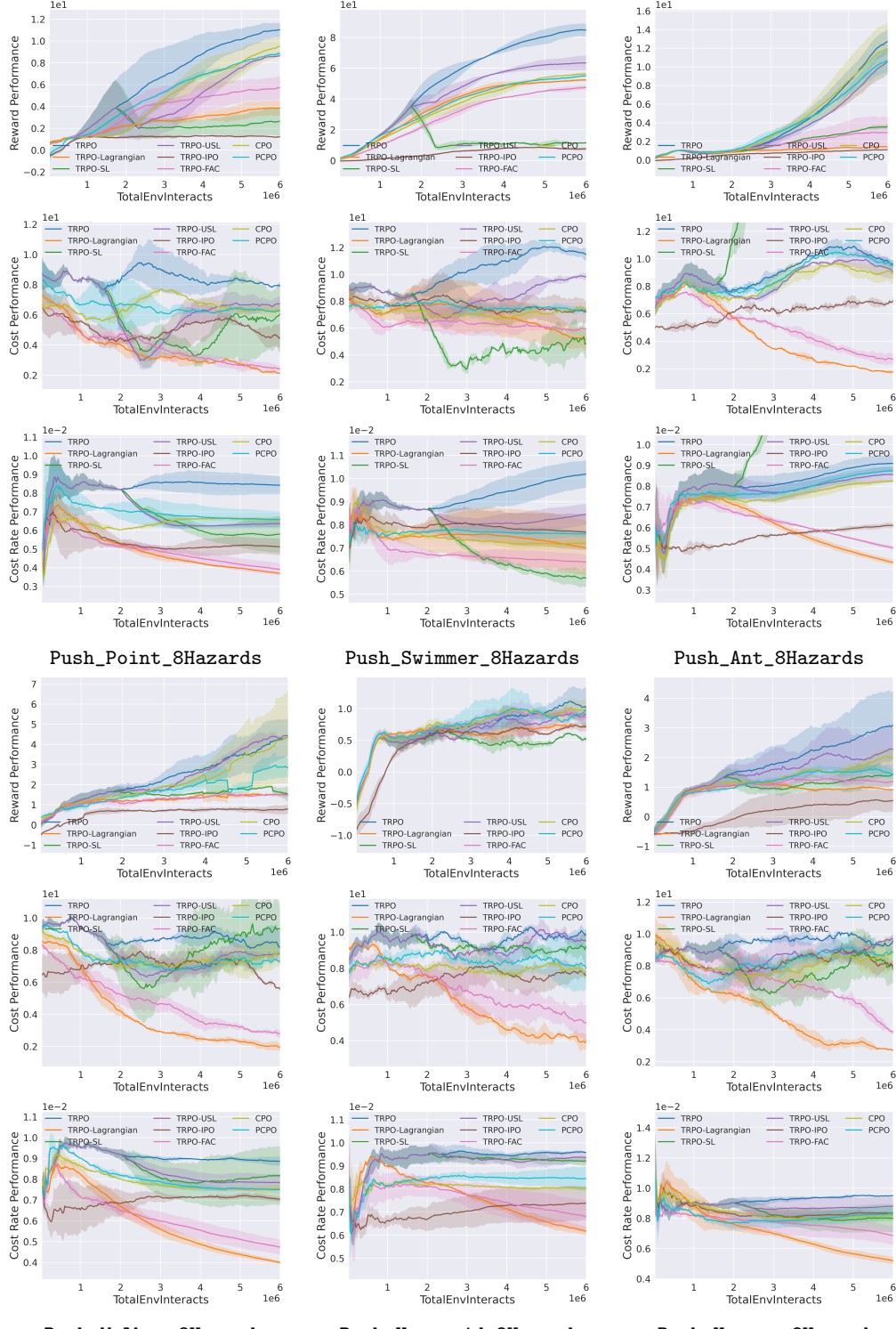

Push_Point_8Hazards    Push_Swimmer_8Hazards    Push_Ant_8Hazards

Push_Walker_8Hazards    Push_Humanoid_8Hazards    Push_Hopper_8Hazards

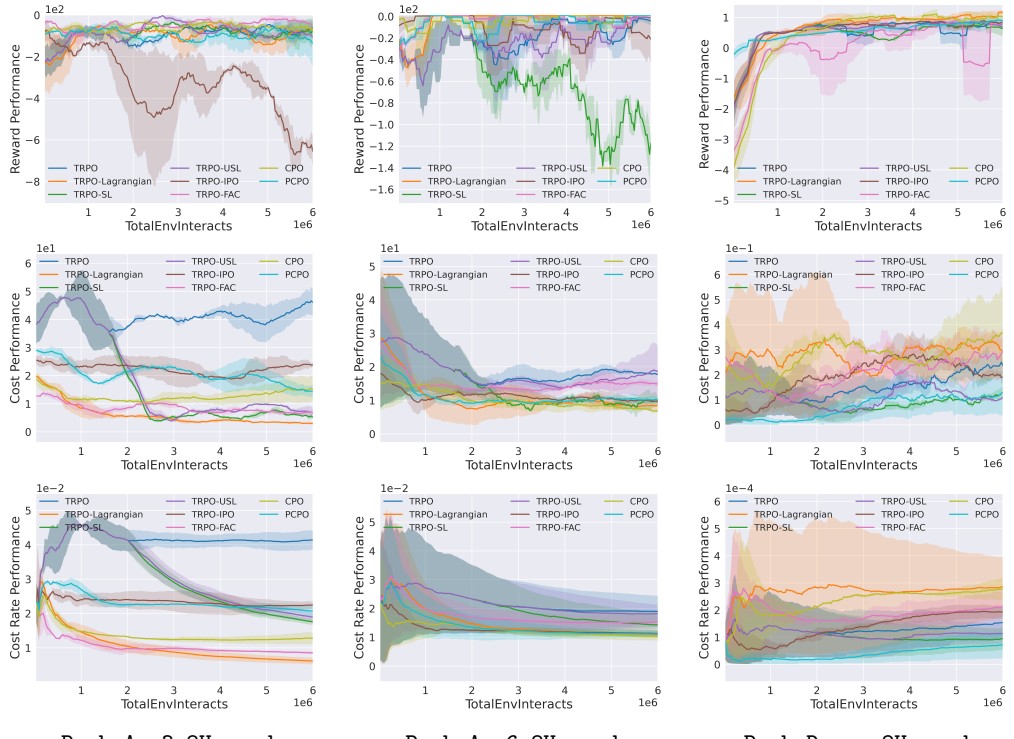

**Push_Arm3_8Hazards**      **Push_Arm6_8Hazards**      **Push_Drone_8Hazards**

Figure 8: `Push_{Robot}_8Hazards`

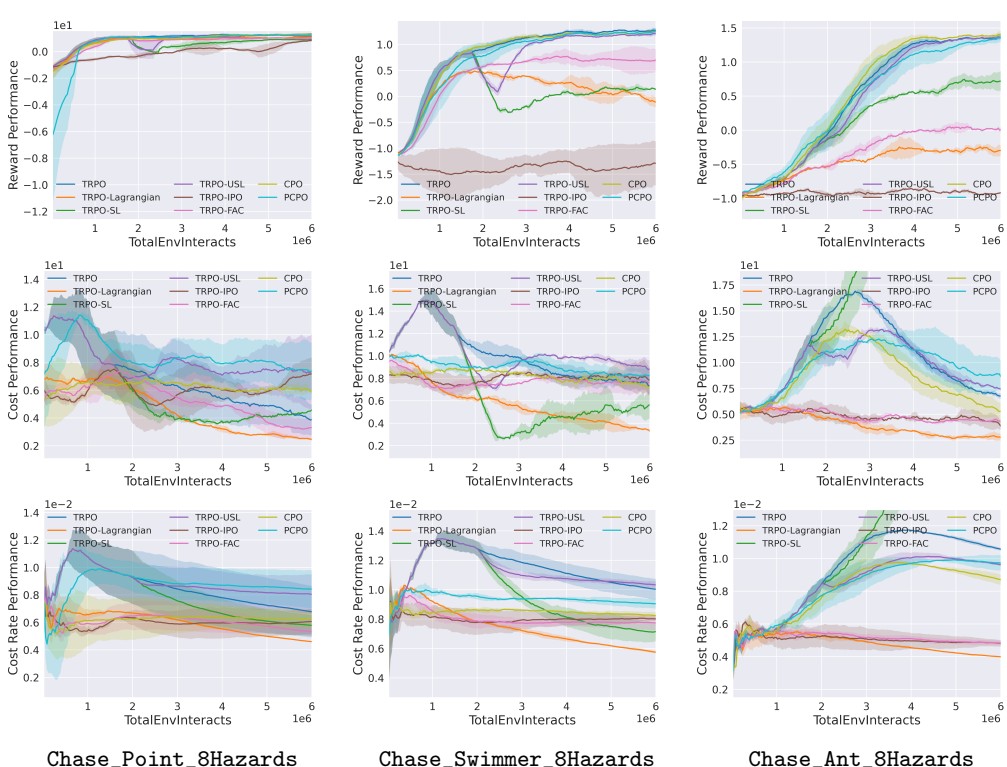

**Chase_Point_8Hazards**      **Chase_Swimmer_8Hazards**      **Chase_Ant_8Hazards**

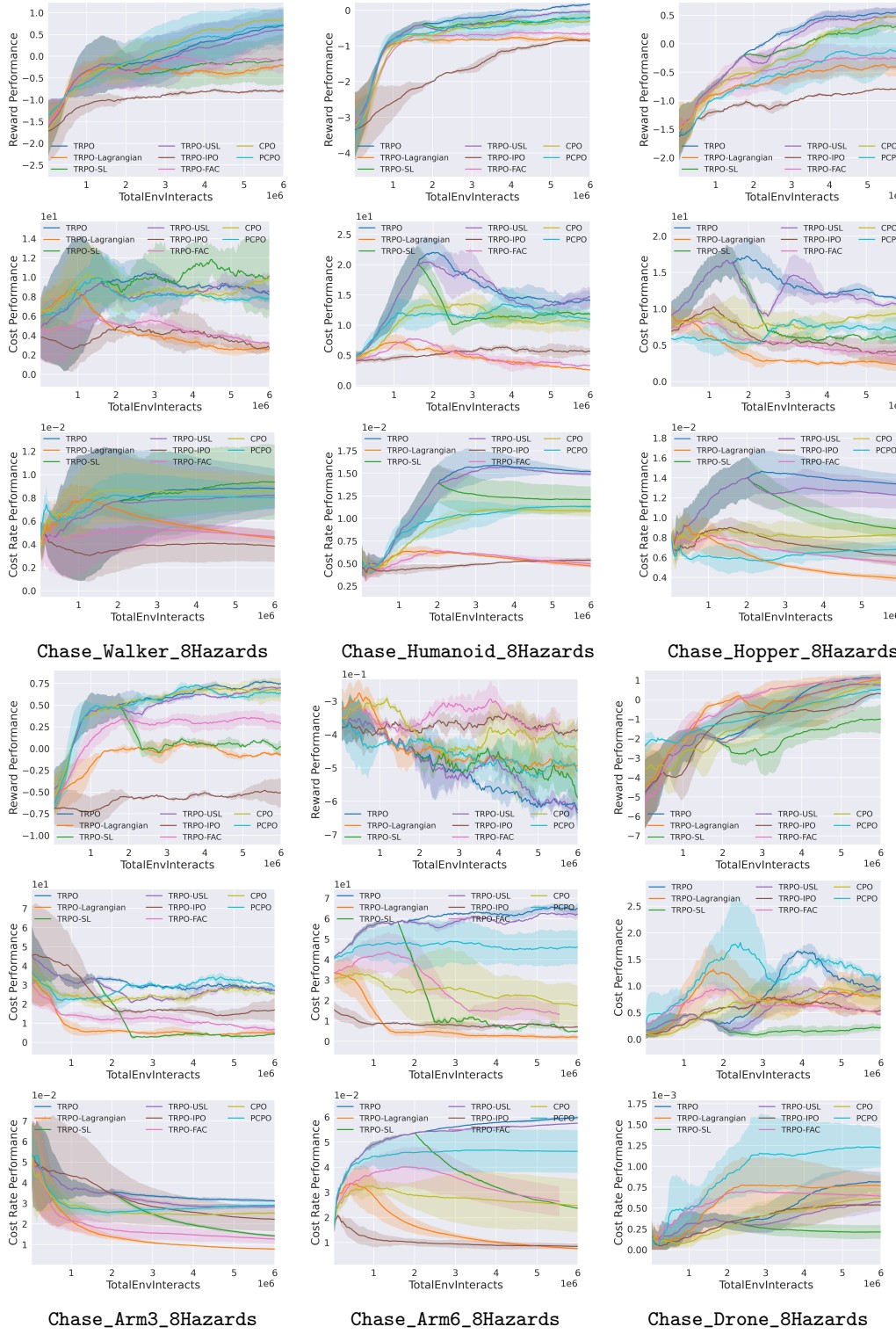

Figure 9: `Chase_{Robot}_8Hazards`

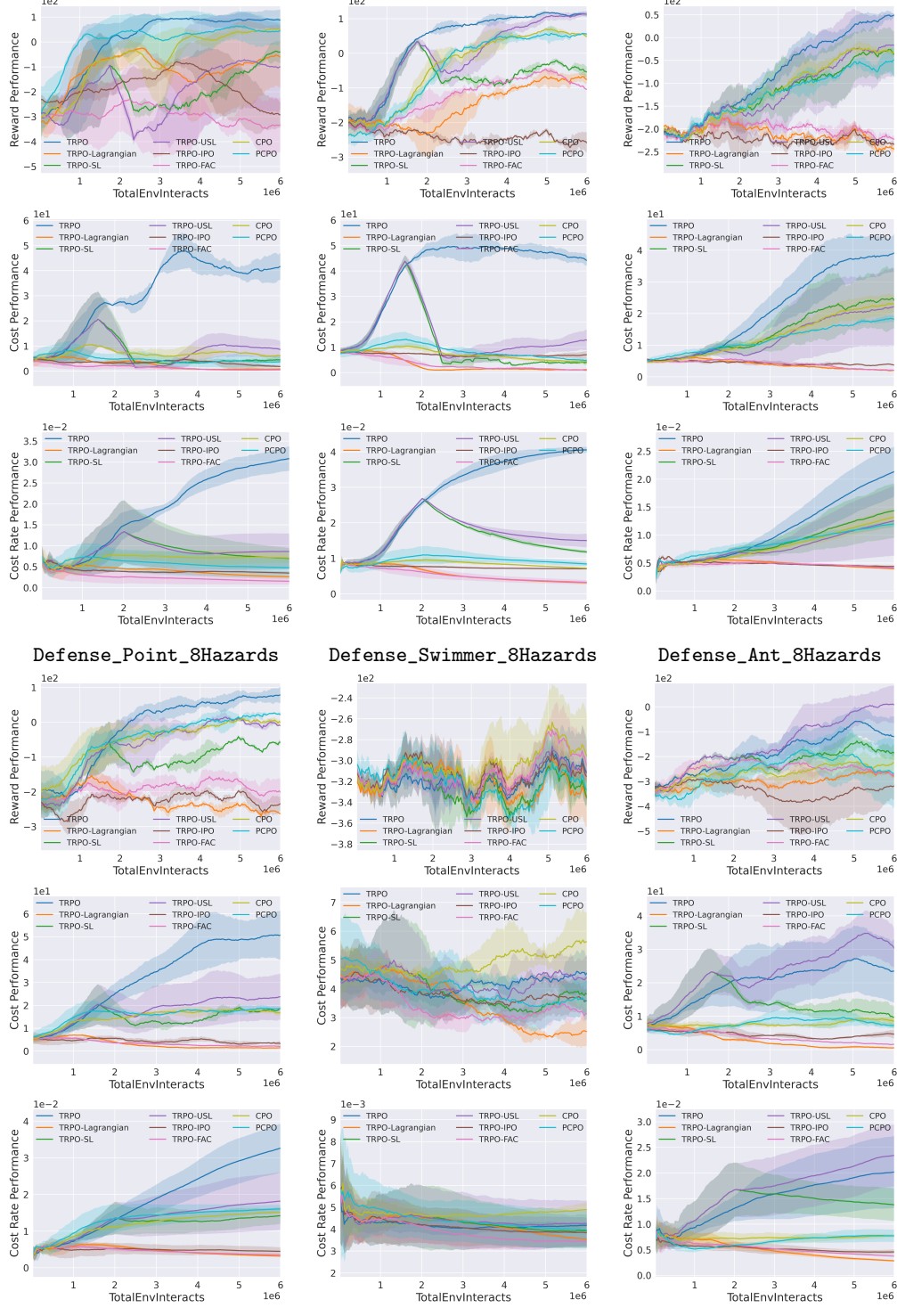

Defense_Point_8Hazards   Defense_Swimmer_8Hazards   Defense_Ant_8Hazards

Defense_Walker_8Hazards   Defense_Humanoid_8Hazards   Defense_Hopper_8Hazards

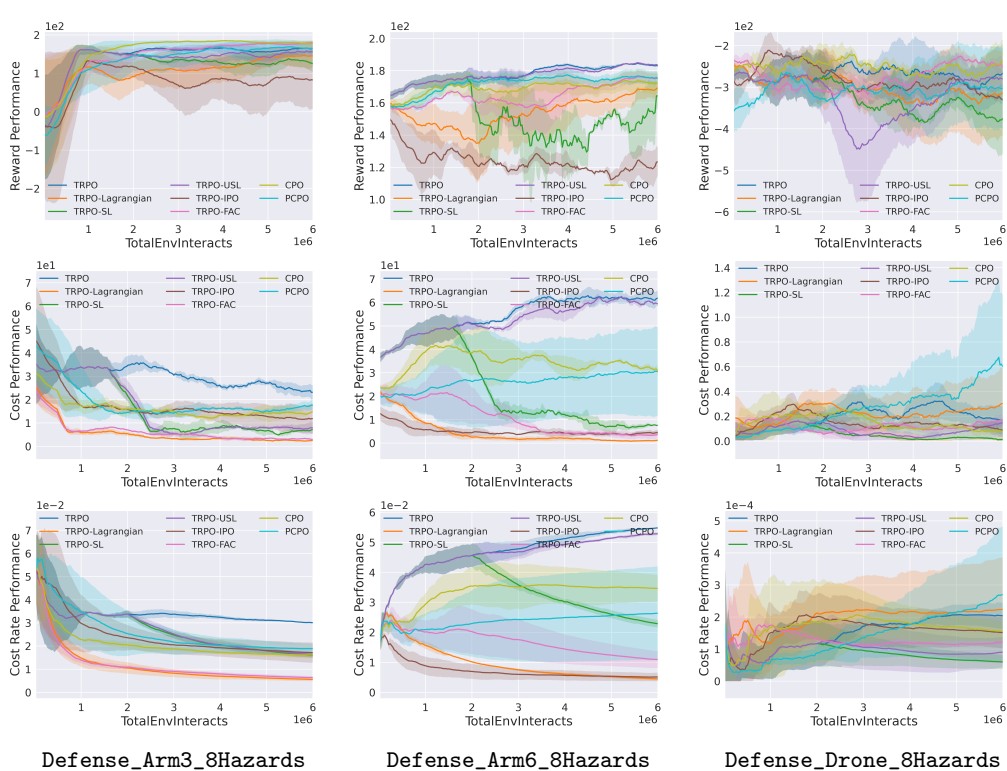

Defense_Arm3_8Hazards       Defense_Arm6_8Hazards       Defense_Drone_8Hazards

Figure 10: `Defense_{Robot}_8Hazards`