# OpenReview forum: "GUARD: A Safe Reinforcement Learning Benchmark"
_NeurIPS.cc/2023/Track/Datasets_and_Benchmarks — Submitted to NeurIPS 2023 Datasets and Benchmarks_

### Official Review · Reviewer_CmEk · 2023-07-19
**Interesting and relevant benchmark for safe RL methods**

**Rating:** 5
**Confidence:** 4
**Correctness:** No correctness related issues.
**Clarity:** The paper is well written.

**Strengths:**

1. Comprehensive benchmark: GUARD serves as a widely encompassing benchmark. It covers a diverse range of RL agents, tasks, and safety constraint specifications, making it applicable to various real-world scenarios and applications. Compared to previous benchmarks, GUARD's comprehensiveness enhances its practicality and versatility.
2. Self-contained implementations: GUARD provides self-contained implementations of state-of-the-art safe RL algorithms. This ensures that other researchers can easily reproduce and directly utilize these algorithms in their work, fostering reproducibility and facilitating experimentation.

**Additional Feedback:**

This paper is more like an experimental report than an academic paper. The authors do not conduct an in-depth analysis of the experimental results and give possible solutions to the problems of Safe RL.

**Documentation:**

The README has sufficient documentation on how to get started with the repository.

**Ethics:**

There is no ethics related issue.

**Limitations:**

The paper may not fully capture the challenges and complexities present in real-world safety-critical applications due to a limited variety of test tasks. Including a more diverse set of tasks would strengthen the benchmark's realism and applicability.

**Opportunities For Improvement:**

1. Lack of clarity in constraint violation analysis: The experiment's absence of visualizing constraint violation thresholds makes it difficult for readers to discern which algorithms violated constraints, which ones did not, and which constrained algorithms achieved the highest rewards without violating constraints. Including such visualizations or clear explanations would enhance the understanding of the results.
2. Insufficient comparison with previous benchmarks: The paper does not adequately highlight the distinctive advantages of GUARD compared to previous safe benchmarks. As a NeurIPS submission, it should present a more in-depth and comprehensive comparison, emphasizing the novel contributions and improvements over existing benchmarks beyond just introducing additional algorithms or test tasks.
3. Addressing challenges and shortcomings: The paper should explicitly outline the remaining challenges in safety reinforcement learning benchmarks and analyze the limitations of current algorithms. By addressing these issues, the authors can demonstrate their understanding of the broader field and propose potential avenues for future research.

**Relation To Prior Work:**

It lacks a comprehensive comparison with previous benchmarks like "A Massively Parallel Benchmark for Safe Dexterous Manipulation"[1] and the "SafePO"[2] project.
[1] https://openreview.net/forum?id=k2Ml8FGtJZp
[2] https://github.com/PKU-Alignment/Safe-Policy-Optimization

**Summary And Contributions:**

The authors present a significant contribution in the field of safe reinforcement learning (RL) with the introduction of GUARD (Generalized Unified SAfe Reinforcement Learning Development Benchmark). The motivation for this work arises from the challenges of applying RL algorithms to safety-critical real-world applications, such as autonomous driving, human-robot interaction, and robot manipulation, where errors are intolerable.

---

> ### Author Response · Authors · 2023-08-24
> **Author Rebuttal**
>
> We thank you for your constructive feedback. Please see our responses below.
>
> ### Opportunities For Improvement
>
> **1** Thank you for your concerns. Due to page limit, we did not include thorough analysis as what would a survey paper have. We will try to include the suggested visualizations in revisions.
>
> **2** In our perspective, the strength of a benchmark toolbox translates into the coverage of algorithms and tasks and easy customization. We will be happy to improve the design if more specific comments are given on desired aspects and capabilities.
>
> **3** Thank you for the suggestions. Same as point 1, we omitted in-depth analysis that would normally appear in a survey paper. We will try to elaborate more on the results in future revisions.
>
> ### Limitations
>
> We agree that real-world robots can be far more complex that what GUARD supports. Ultimately, safe RL algorithms should handle high-fidelity agents and tasks, but that goal takes many steps to reach. The robot complexity (as high as 10D) and task complexity (interactive ones such as chase and defense) in GUARD are already over the limit of what SOTA safe RL methods can handle. We believe that GUARD represents an important step towards the ultimate goal at the current time.
>
> ### Relation To Prior Work
>
> Thank you for pointing out. We will cite and discuss both citations in our revisions as:
> “Safety Gymnasium [cite] improves Safety Gym by optimizing software dependencies and supports four navigation tasks. SafePO [2] integrates Safety Gymnasium with Safety DexterousHands [1], another safe RL benchmark with a focus on high-fidelity manipulation tasks.”
>
> ### Additional Feedback
>
> We believe that we prepared our manuscript with the same high standard as any academic paper in content, format, and presentation. The focus of a benchmark paper is the description of the benchmark as a tool. Our paper provides in-depth details of GUARD to fulfill that purpose. Experimental results mainly showcase the capability of GUARD as a tool, hence we provided a relatively succinct analysis due to the page limit. We are happy to extend the analysis (see the response to reviewer v1GQ) as a paper revision. One may provide in-depth result analysis and possible solutions in a survey paper, where results can be efficiently generated using our GUARD platform.

---

### Official Review · Reviewer_nmvU · 2023-07-20
**A great benchmark with extensive algorithm implementations and convincing results, but not satisfying in terms of tasks and environments.**

**Rating:** 4
**Confidence:** 3
**Correctness:** I believe most contents presented in …

**Strengths:**

+ I'm not an expert in the field of safe RL, but it seems that the algorithm implementation in this benchmark is extensive. Also, the authors present thorough evaluation results (i.e., learning curves) and corresponding implementation details across a wide range of environments. These results seem to be reproducible and convincing.

+ The authors provide end-to-end RL training implementations with many robot types. They also present detailed configuration options in the appendix.

**Additional Feedback:**

N/A

**Clarity:**

This paper follows the logic of a classic benchmark paper in the field of RL. The presentation is clear and easy-to-follow.

**Documentation:**

I think details in the appendix together with the release code in GitHub repo are sufficient to reproduce the experiment results.

**Ethics:**

No ethics review required.

**Limitations:**

The limitations are well-addressed in the above section.

**Opportunities For Improvement:**

+ While providing reference implementations of safe RL algorithms is great, I think another important property of a good benchmark is that users can implement their own ideas in this library. For algorithms, yes, the codebase is based on spinningup and basically single-file. Users can easily implement new algorithms via modifying existing ones. However, for tasks and constraint options, I found that the released implementation is quite messy. Could the authors explain in detail what should the user do if, e.g., he/she wants to set up two walls beside a robot, and give a cost of 1 if the robot hits walls and 0 otherwise? It seems that it is extremely hard to add a new robot type or safety constraint. The consequence is that users can only use the provided tasks and constraints, which may be problematic due to the potential "bias" of the authors (I mean the constraint types and tasks are inevitably limited).

+ The authors claim that the benchmark features a modularized design, but I don't think the codebase is generally modularized as I explained in the previous point. I think the authors should consider further modularization of the environment code.

+ This is a minor point, but the appendix is literally too long and too large. I think it does not generally help if the authors present detailed task names, action and observation spaces in the appendix. They could be placed in a link to the documentation site. It occupies too much spaces for truly useful contents. Besides, try to use the pdf format for inserting images.

**Relation To Prior Work:**

The authors have clarified that this benchmark is developed based upon several previous libraries. This work is basically an extension of previous works.

**Summary And Contributions:**

This paper proposes a benchmark on safe reinforcement learning. The authors combine existing library/implementations, namely safetyGym and spinningup, and extend them with state-of-the-art safe RL algorithms. They also integrate many robot types (based on the MuJoCo simulator), new tasks, and configurable safety constraints into their environments. They evaluate the implemented algorithms and present learning curves in the paper as common benchmark papers do. This benchmark could possibly be used as the reference implementation or baselines in futural safe RL research.

---

> ### Author Response · Authors · 2023-08-24
> **Author Rebuttal**
>
> We thank you for your feedback. Please see our responses below.
>
> ### Opportunities For Improvement
>
> **I1** Thank you for the comment. We included a tutorial for adding new tasks in the repo readme (Quick Start - 1. Environment Configuration) where the users can select among existing tasks, robots, targets, obstacles, and constraints. Walls can be added as non-trespassable obstacles easily. On the other hand, the difficulty of adding new robot and constraint types come from MuJoCo programming with no easy way to bypass it. To help with deeper user customization, we will summarize the general procedure for adding new robots and constraints, and refer to external links for reference in the online repo readme file.
>
> **I2** Thank you for the suggestion. We will try to redesign the codebase to enable modularized configurations of robot types and constraint types in future developments.
>
> **I3** Thank you for the suggestions. We will adapt our future revisions accordingly.

---

> > ### Comment · Reviewer_nmvU · 2023-08-29
> >
> > I thank the authors for their responses. However, they did not address my core concerns, and I decide to maintain my current rating.

---

> > > ### Author Response · Authors · 2023-08-31
> > >
> > > We would highly appreciate it if the reviewer pointed out the core concerns that are not addressed. Thank you!

---

### Official Review · Reviewer_v1GQ · 2023-07-22
**Reviews for GUARD**

**Rating:** 7
**Confidence:** 4

**Strengths:**

* Diverse agent and task coverage: The paper provides an extensive list of 11 robot agents, 7 task specifications, and 8 safety constraint options, making GUARD a versatile and practical benchmark for evaluating safe RL algorithms.

* Comprehensive Algorithm Comparison: The authors include eight state-of-the-art safe RL algorithms for comparison on different agents/tasks/safety constraints, validating the usability of the benchmark.

**Additional Feedback:**

None

**Clarity:**

The paper is generally well-written and structured. Providing more explanations and examples, and a summary of the algorithm performance comparisons could improve clarity.

**Correctness:**

The paper appears to be technically sound, providing a well-defined benchmark framework.

**Documentation:**

Nice documentation for both the tasks and benchmark algorithm.

**Limitations:**

The paper does not explicitly mention limitations of the proposed GUARD benchmark in the paper. It would be nice to include an explicit discussion for the limitations of this work and future directions to tackle these limitations.

**Opportunities For Improvement:**

* Include more discussion about the algorithm comparison results: While the paper introduces extensive algorithm comparison results in Figure 4, more discussions about the performance of the algorithm would enhance its practical value. For example, including a summary of key findings, highlighting strengths and weaknesses of each algorithm under different scenarios, would enable readers to draw more meaningful conclusions.

* Supporting Movable and Multiple Targets for Goal Task: The paper mentions GUARD's support for Immovable targets in goal-reaching tasks. Elaborating on the specific challenges faced for movable and multi- targets, and potential extending GUARD's support for Movable and Multiple Targets would be valuable.

**Relation To Prior Work:**

The paper adequately establishes the relation to prior work, mentioning existing safe RL benchmarks and the contribution of GUARD.

**Summary And Contributions:**

The paper introduces a comprehensive benchmark for safe reinforcement learning (RL) algorithms. It extends existing safe RL benchmarks by incorporating a wide range of agents, tasks, and safety constraints.

---

> ### Author Response · Authors · 2023-08-24
> **Author Rebuttal**
>
> We highly appreciate your constructive feedback. Please see our responses below.
>
> ### Opportunities For Improvement
>
> **I1** Thank you for the suggestions. Due to the page limit, we are succinct in performance analysis in section 6. We will try to add more discussions in the paper revision. Here is a tentative modified version of the analysis:
>
> We have conducted an in-depth examination of several reinforcement learning algorithms, including Unconstrained Methods (TRPO), Constrained Methods (TRPO-IPO, TRPO-Lagrangian, PCPO, CPO, TRPO-FAC), and Hierarchical Methods (USL and Safelayer). Our analysis has yielded important insights into their respective performances, which we will outline below.
>
> Unconstrained Methods (TRPO):
>
> Pros: TRPO-based methods demonstrate commendable high reward performance.
>
> Cons: However, they do not factor in cost considerations, which may limit their applicability in scenarios where cost-efficiency is a critical concern.
>
> Constrained Methods (TRPO-IPO, TRPO-Lagrangian, PCPO, CPO, TRPO-FAC):
>
> Pros: Constrained methods take cost considerations into account from the beginning of training, making them a favorable choice when cost-efficiency is a primary objective.
>
> Cons: Nevertheless, they tend to exhibit slower reward increases when compared to unconstrained TRPO.
>
> Specific Notes on Constrained Methods:
>
> TRPO-IPO: It is noteworthy that TRPO-IPO exhibits the lowest reward performance among the algorithms considered, suggesting that it may not be the optimal choice in scenarios where maximizing rewards is paramount.
>
> TRPO-Lagrangian: TRPO-Lagrangian excels in terms of cost-effectiveness across most tasks, but it faces challenges when applied to the Drone robot.
>
> Hierarchical Methods (USL and Safelayer):
>
> Pros: Hierarchical methods offer an advantage by being easily integrable into exit methods as an additional component of the environment, enhancing safety considerations.
>
> Cons: However, they require a warm-up period to collect data for better safety layer performance. Additionally, reward performance tends to decline post-warm-up.
>
> Specific Notes on Hierarchical Methods:
>
> Safelayer: Safelayer's performance stands out with the Drone robot, achieving exceptional cost-effectiveness without significant sacrifice in reward performance. However, it shows limitations on the Ant robot, potentially leading to increased overall costs in that specific scenario.
>
> **I2** Thank you for your suggestions. Those are definitely useful extensions, and we will include them in future developments.
>
> ### Limitations
>
> Thank you for the suggestion. We will add the following discussion to the paper revision:
>
> The GUARD benchmark paper indeed faces the limitation of having a limited set of evaluation metrics. This poses challenges in providing a comprehensive assessment of reinforcement learning agents' performance. To address this limitation, it is crucial to suggest additional or alternative evaluation metrics that can offer a more holistic view of an agent's capabilities. By expanding the range of metrics, we can better gauge an agent's proficiency in handling various aspects of the task, ultimately leading to a more robust benchmark.
>
> Another limitation highlighted in the paper is the reliance on the deprecated mujoco_py library. In the rapidly evolving field of reinforcement learning, staying up-to-date with the latest software libraries is essential. To overcome this limitation, it is imperative to update the Mujoco interface to be compatible with the new library version. Ensuring compatibility with the latest tools and libraries will help maintain the benchmark's relevance and usability for researchers in the field.
>
> The GUARD benchmark currently focuses solely on on-policy reinforcement learning algorithms. While this is valuable for certain applications, it limits the benchmark's versatility and usefulness across a broader range of reinforcement learning strategies. To mitigate this limitation, an essential future direction is to extend the algorithms library to include a more diverse set of reinforcement learning strategies, encompassing both on-policy and off-policy algorithms. This expansion will make the benchmark more comprehensive and relevant to a wider audience of researchers.
>
> The instability of simulation when subjected to extreme actions for specific robots is another limitation worth addressing. To tackle this challenge, it is essential to fine-tune the parameters for the robots and the environment. This adjustment can help ensure stability and robustness in the simulation, providing a more accurate representation of real-world scenarios. Moreover, it's important to document these parameter adjustments to maintain transparency and reproducibility in the benchmarking process.

---

### Official Review · Reviewer_xtKP · 2023-07-25
**a benchmark about safe reinforcement learning**

**Rating:** 4
**Confidence:** 4
**Correctness:** Yes
**Clarity:** Yes

**Strengths:**

Strengths:

- The GUARD considers an important problem of facilitating fair benchmarking and comparison of safe RL algorithms. A unified platform can enable more rigorous evaluation.
- The work attempts to provide a generalized framework supporting a diverse range of agents, tasks, and safety constraints, which enhances flexibility.
- GUARD implements several safe RL algorithms to facilitate quick benchmarking.
- The benchmark results provide baseline performance analysis on different test cases, elucidating general trends.

Weaknesses:

- Upon attempted replication, I regret to report that the codebase for this platform available on GitHub is presently non-functional and fails to run.
- This work lacks novelty, as GUARD appears to be an incremental extension of prior work without introducing substantial new capabilities beyond existing platforms.
- There already exist corresponding high-quality algorithm libraries and environment libraries in this field, which the authors appear to have overlooked in the context of prior work and the novelty of contributions.
- The benchmark experiments offer limited insights into the nuances of different algorithms. More thorough analysis and ablation studies could reveal deeper trade-offs.
- The contributions are not adequately justified or framed to demonstrate significance for the Safe RL community.

In conclusion, GUARD aims to address the significant challenge of facilitating standardized benchmarking for safe reinforcement learning algorithms, but this work has critical shortcomings in terms of novelty, benchmark analysis, implementation details, technical depth, and framing of contributions that require addressing before acceptance. I hope these strengths and weaknesses provide constructive feedback to improve the work.

**Additional Feedback:**

No

**Documentation:**

Yes

**Limitations:**

Limitations:

1. Attempting to run the codebase per the README instructions resulted in over 4 hours spent installing dependencies yet still encountering errors regarding mujoco_py, preventing execution. This reveals limitations of the repository's facilitation of research community reuse, as reviewers cannot verify results and researchers will likely struggle to leverage it.
2. GUARD's reliance on the officially deprecated mujoco_py exposes limitations in considering long-term community support. As a newly proposed library, adopting superseded dependencies risks hampering sustainability and continued relevance. The authors should evaluate platform longevity when selecting component technologies.
3. I notice the codebase inherits from SpinningUp and each algorithm is implemented in a self-contained manner. However, from a software engineering standpoint, such modularity does not necessarily enhance extensibility.
4. The 11 agents included appear to cover only a narrow fraction of real-world robot diversity. Most real applications would involve robotic systems with substantially higher degrees of freedom, more complex dynamics and interactions, and less idealized movements than the relatively simple agents implemented. For instance, real-world robots exhibit far greater heterogeneity in locomotion and manipulation capabilities.
5. The safety constraints introduced by the authors do not comprehensively cover the diverse requirements necessary for real-world applications. Practical scenarios demand modeling a wide variety of constraints, such as joint limits to prevent damage, restrictions on robot speed and forces for safe collaboration, and many others. The hazards, ghosts, and simple contact dynamics currently implemented represent only a fraction of the constraints needed to emulate complex, nuanced real-world conditions.
6. The authors assert that GUARD provides a "unified benchmarking platform with comprehensive coverage of safe RL algorithms." However, when examining the algorithms implemented, GUARD incorporates only a subset of prevailing safe RL techniques. Many important categories are absent, including model-based methods, active safe learning schemes, and various multi-objective algorithms. The covered algorithms reflect only a fraction of the diverse methodologies employed for safe RL. Hardly "comprehensive", GUARD's algorithm implementations are limited primarily to RL trust-region methods and simple Lagrangian approaches.
7. GUARD covers only basic constraints, while comprehensive coverage necessitates incorporating much more complex and diverse safety requirements. The current constraints are limited to idealized spheres and circles, lacking fidelity for real-world applicability. Significant expansion of constraint types is required before one can claim comprehensive scope based merely on incremental additions to Safety Gym.
8. The proclaimed Generalized and Unified attributes of GUARD are inadequately embodied based on the limitations highlighted. This directly impacts the practical significance of the work. The authors must better validate that contributions fulfill stated goals to demonstrate meaningful advancements.
9. In the benchmark results, the authors showcase the Push task for three agents - Arm3, Arm6, and Drone - with 8 hazards as constraints. However, the reward learning curves for these cases exhibit no meaningful improvement over time and display high volatility. The failure to attain clear reward increases despite training runs suggests possible issues in task formulation, algorithm tuning, or other factors. Moreover, the authors showcase the Chase task for the Arm6 agent avoiding 8 hazards. However, the reward learning curve for this case exhibits a decreasing trend rather than the expected increasing performance. This mismatch suggests potential issues in the task formulation or the algorithm itself.

**Opportunities For Improvement:**

Questions:

- The paper mentions GUARD significantly extends existing platforms in agents, tasks, and constraints. Could the authors provide more specifics on the quantitative extent of improvements versus current state-of-the-art benchmarks?
- The authors have not discussed or cited major existing safe reinforcement learning libraries such as Safe-Control-Gym, and Safety Gymnasium when presenting the motivation and goals for GUARD. Is the lack of comparison against predecessors indicative of insufficient literature review? Or do the authors believe those platforms exhibit capabilities markedly weaker than GUARD?
- The availability of artifacts is an important requirement, considering the Gymnasium's documentation contributes a lot to the RL community. Is the codebase open source with proper documentation to enable verification by external researchers?
- Given the setting of ghosts. Doesn't RL depend on Markovian environments and reward design, which seems at odds with adapting unpredictable actions?
- In Figure 4, why do subfigures (f) to (h) evaluate an unexplained "point robot" on various benchmark tasks?

**Relation To Prior Work:**

Yes

**Summary And Contributions:**

GUARD claims (1) A generalized platform supporting diverse agents, tasks, and constraints; (2) Unified implementation covers 8 major safe RL algorithms; (3) Self-contained architecture enables simple usage and understanding; (4) Range of predefined benchmark suites and results; (5) Experiments analyzing for algorithm behaviors on tasks with varying properties.

---

> ### Author Response · Authors · 2023-08-24
> **Author Rebuttal**
>
> ### Weaknesses
>
> **W1** We are sorry to hear that. The released codebase is presently functional which is verified by at least one researcher in the community. Please feel free to post technical issues under our repository, and we will respond ASAP like what we did here: https://github.com/intelligent-control-lab/guard/issues/1
>
> **W2** The novelty of GUARD is to provide a unified framework for fast algorithm testing on various tasks to save efforts. GUARD is novel with the most agent types, task types, and safety constraint types so far with easy-to-modify code structures. The coverage of safety constraints is especially substantial compared to existing work. Please enlighten us of any specific missing capabilities vital to safe RL, and we will be happy to discuss them.
>
> **W3** We have included a comprehensive review of existing high-quality algorithms and environment libraries in our related work. Please let us know of any specific library that is missing and we’ll be happy to include.
>
> **W4** We have included an analysis of different algorithms in section 6 based on our extensive testing. The focus of this paper is to describe the testing capabilities of GUARD (see sections 4 and 5), where the in-depth interpretation of results is more suited for a survey paper.
>
> **W5** We would appreciate it if specific deficiencies or unaddressed issues were provided.
>
> ### Questions
>
> **Q1** We listed limitations of current SOTA benchmarks in terms of number of supported algorithms and tasks in section 2 (e.g., Safety Gym with 3 safe RL methods, Bullet-Safety-GYm supporting 4 methods, etc.). The improvement is evident given our quantities in section 1.
>
> **Q2** We did discuss about Safe-Control-Gym in section 2. Thanks for bringing up Safety Gymnasium. We did look at this repo, and found that it has limited tasks in both quantity (4 vs. 7 in ours) and lacks complex tasks such as defense and chase. We will be happy to cite Safety Gymnasium.
>
> **Q3** We agree that Gymnasium has a good documentation page. In our case, GUARD is open source with full documentation in the submitted paper which is available online. We will be happy to convert the paper into a documentation page in the future.
>
> **Q4** The movement of the ghosts depends on the agent state and, hence is captured by the Markov decision process. This is described in Appendix A.5.3.
>
> **Q5** Point robot is a standard and commonly known agent type from Safety Gym.
>
> ### Limitations
>
> **L1** Mujoco_py is not included in our release, hence the error encountered is mostly likely specific to the actual user system. We’ll be happy to help if the issue is posted under our GitHub repository. Our codebase is functional as at least one researcher in the community was able to run our codebase after resolving a minor issue (see our git issue tab).
>
> **L2** We aim to enhance the user experience by offering more versatile interfaces for running a wide array of RL algorithms across different tasks. The simulation component of this benchmark was finalized before the deprecation of mujoco_py, prompting our decision to temporarily transition to the stable version of the Mujoco interface. In the future, we will preserve the current benchmark interfaces for users and adapt to the new Mujoco library.
>
> **L3** Shared pipelines need to accommodate the train/test procedures of all RL methods. Any change to one method might require a change to the pipeline, which is extra overhead to the originally succinct RL methods. Being self-contained, GUARD allows fast customization and extension. GUARD achieves an appropriate balance between shared structure and customizability for research.
>
> **L4** We agree that real-world robots can be far more complex. Ultimately, safe RL algorithms should handle real-world cases, but that takes many steps to reach. The robot complexity (as high as 10D) and task complexity (interactive ones such as chase and defense) in GUARD are already over the limit of SOTA methods. We believe that GUARD represents an important step towards the ultimate goal.
>
> **L5** It takes steps to achieve such diversity, and no platform is able to support all real-world cases. GUARD already surpasses existing platforms substantially in terms of safety constraints, poses great challenges for safe RL algorithms, and hence is a meaningful step.
>
> **L6** In section 4.1, we mentioned that GUARD only considers model-free approaches which depend on the least assumptions. Model-based methods often depend heavily on assumptions. Regarding active safe learning schemes and various multi-objective algorithms, we will be happy to discuss if specific methods are provided.
>
> **L7** Duplicate of L5.
>
> **L8** We will be happy to discuss more if actionable feedback is provided.
>
> **L9** RL algorithms are naturally volatile in complex tasks. Arm and drone cases with 8 harzrads are very difficult for safe RL. We include them to expose shortcomings and motivate new advancements. Mixed algorithm performance is expected here.

---

> > ### Comment · Reviewer_xtKP · 2023-08-28
> > **Re: Author Rebuttal**
> >
> > The author's response seemed sloppy and did not address my core concerns, and I decided to maintain my current grade.

---

> > > ### Author Response · Authors · 2023-08-31
> > >
> > > Thank you for your response. We would highly appreciate it if the reviewer pointed out the core concerns that are not addressed, with advice on how we might improve. Thanks again!

---

### Decision · Program_Chairs · 2023-09-22

**Decision:**

Reject

**Comment:**

On the positive side, the reviewers believe that GUARD in principle fills an important gap.

However, the reviewers also are strongly concerned about both the novelty of the challenge and the accessibility of the codebase.
One reviewer in particular attempted to run the code and ran into practical issues, preventing them from replicating the results.

On this basis I recommend the rejection of the paper and submission to a future venue.